# Tuning orbital-selective phase transitions in a two-dimensional Hund's correlated system

Eun Kyo Ko[1,2,7], Sungsoo Hahn [2,3,7], Changhee Sohn [4], Sangmin Lee[5], Seung-Sup B. Lee [2,6], Byungmin Sohn [1,2], Jeong Rae Kim[1,2], Jaeseok Son [1,2], Jeongkeun Song[1,2], Youngdo Kim[1,2], Donghan Kim[1,2], Miyoung Kim[5], Choong H. Kim [1,2] ✉, Changyoung Kim[1,2] ✉ & Tae Won Noh [1,2] ✉

Hund's rule coupling ($J$) has attracted much attention recently for its role in the description of the novel quantum phases of multi-orbital materials. Depending on the orbital occupancy, $J$ can lead to various intriguing phases. However, experimental confirmation of the orbital occupancy dependency has been difficult as controlling the orbital degrees of freedom normally accompanies chemical inhomogeneities. Here, we demonstrate a method to investigate the role of orbital occupancy in $J$ related phenomena without inducing inhomogeneities. By growing $SrRuO_3$ monolayers on various substrates with symmetry-preserving interlayers, we gradually tune the crystal field splitting and thus the orbital degeneracy of the Ru $t_{2g}$ orbitals. It effectively varies the orbital occupancies of two-dimensional (2D) ruthenates. Via in-situ angle-resolved photoemission spectroscopy, we observe a progressive metal-insulator transition (MIT). It is found that the MIT occurs with orbital differentiation: concurrent opening of a band insulating gap in the $d_{xy}$ band and a Mott gap in the $d_{xz/yz}$ bands. Our study provides an effective experimental method for investigation of orbital-selective phenomena in multi-orbital materials.

Mott physics with strong electron correlations due to on-site Coulomb repulsion ($U$) have been a central paradigm in condensed matter[1,2]. Recently, Hund's metal, a new type of strongly correlated material, was proposed, for which Hund's rule coupling ($J$) dominantly drives the electron correlations rather than $U$[3–12]. In such a system, the roles of $J$ are highly dependent on the number of electrons/orbitals, so a variety of physical phenomena depend strongly on the orbital occupancy. In particular, rich phases with orbital differentiations have been suggested theoretically for Hund's correlated system[3,13,14]. Since the energy scale of $J$ is smaller than that of $U$

for most multi-orbital materials, the associated phase transitions can be achieved by controlling an energy scale much smaller than that in Mott physics[3,13,14].

Despite extensive theoretical interest[3,6,7,10,13–18], observations of Hund-driven phase transitions have remained challenging because direct control of the $J$ value is experimentally difficult as it is usually determined by atomic physics. On the other hand, an alternative but indirect approach to control orbital occupancy is possible via doping and/or substitution with different chemical elements[19,20]. During such experiments, however, numerous defects are easily

[1]Center for Correlated Electron Systems, Institute for Basic Science (IBS), Seoul 08826, Republic of Korea. [2]Department of Physics and Astronomy, Seoul National University, Seoul 08826, Republic of Korea. [3] Research Institute of Basic Sciences (RIBS), Seoul National University, Seoul 08826, Republic of Korea. [4]Department of Physics, Ulsan National Institute of Science and Technology, Ulsan, Republic of Korea. [5]Department of Materials Science and Engineering and Research Institute of Advanced Materials, Seoul National University, Seoul 08826, Republic of Korea. [6]Center for Theoretical Physics, Seoul National University, Seoul 08826, Republic of Korea. [7]These authors contributed equally: Eun Kyo Ko, Sungsoo Hahn. ✉e-mail: chkim82@snu.ac.kr; changyoung@snu.ac.kr; twnoh@snu.ac.kr

formed due to the random distribution of chemical elements. These inhomogeneity problems place serious limitations on experimental investigations of Hund's systems. Therefore, precise control of orbital occupancy without random chemical distribution is the key for experimental investigation of phase transitions in Hund's systems.

We propose that crystal field splitting can be controlled with a suitable experimental approach to observation of phase transitions in Hund's systems with negligible impurity problems. In cubic perovskite oxides ($a$-lattice constant ($a$) = $c$-lattice constant ($c$)), the five $d$-orbitals of transition metal elements are split into $t_{2g}$ and $e_g$ levels due to the oxygen octahedral environment (Fig. 1a). When the oxygen octahedron becomes distorted, the orbitals experience further tetragonal crystal field splitting ($\Delta_t$). For instance, the three $t_{2g}$ orbitals split into $d_{xy}$ and $d_{xz/yz}$ levels if the oxygen octahedron is elongated or compressed. Such variation of $\Delta_t$ can tune the orbital occupancy of the $d$-orbitals without chemical doping in partially-filled $d$-electron systems[21]. There have been previous experimental studies that demonstrated the manipulation of orbital polarization[22–24]. They include the charge transfer at the cuprate-manganate interfaces[22], metal-insulator transition in $VO_2$[23], and nickelate-cuprate heterostructures[24]. However, the impact of $J$ in such orbital polarization changes has not been considered. Here, our objective is to explore the interplay between $\Delta_t$ and $J$ by controlling the orbital polarization in widely accepted Hund's system.

In this study, we investigated tuning of orbital occupancy in SrRuO₃ (SRO) ultrathin films by artificially controlling the crystal field splitting. It should be noted that Sr₂RuO₄ is well-known as Hund's metal, so 2D limit of SRO can provide insight on the Hund's physics[7]. In the bulk SRO, $\Delta_t$ is small, and the nearly degenerate $d_{xy/xz/yz}$ orbitals are partially-filled with 4 electrons (3-orbital/4-electron system). As

shown in Fig. 1b, with an increase in $\Delta_t$, the $d_{xy}$ ($d_{xz/yz}$) band moves downward (upward), leading to a redistribution of electrons among the three $t_{2g}$ orbitals. When the energy level of the $d_{xy}$ band is lower than the Fermi energy ($E_F$), the $d_{xy}$ ($d_{xz/yz}$) band becomes fully-filled (half-filled). Then, SRO will behave effectively as a 2-orbital/2-electron system. This crystal field splitting effect (i.e. with negligible $U/D$) makes $d_{xy}$ level lower than that of $d_{xz/yz}$. When $U/D$ is large ($D$ is the half-bandwidth), the $d_{xz/yz}$ bands can further experience a Mott transition[14], as schematically shown in the last band configuration of Fig. 1b. The relative position of $d_{xz/yz}$ can be a good indication of how $U$ and the crystal field play roles in determining electronic structures in of SRO.

For a comprehensive understanding of the orbital-selective phase transitions, we should pay much attention to the roles of $J$[14]. Figure 1c schematically displays how the phase transitions occur in SRO for two different $J$ values. With $J = 0$, it can be challenging to detect the Mott transition induced by the crystal field because it only happens in a small range of $U$. On the other hand, $J$ facilitates such transition in a much wider interaction range. Therefore, the system with a sizable $J$ experiences orbital-selective phase transitions as $\Delta_t$ varies. For small $U/D$, increasing $\Delta_t$ induces a transition from a metallic state into another metallic state with a gap in the $d_{xy}$ band while, for a large $U/D$, $\Delta_t$ causes a phase transition from a Hund's metal to an insulating state with band ($d_{xy}$) + Mott ($d_{xz/yz}$) gaps. Note that for small $\Delta_t$, the critical $U$ value for opening the Mott gaps ($U_c$) increases when we take $J$ into account (dashed arrow (1) in Fig. 1c). On the other hand, with large $\Delta_t$, $U_c$ decreases with sizable $J$ (dashed arrow (2)). This variable nature of $J$ makes the phase transition with orbital differentiation intriguing[3,14]. Therefore, our experimental approach with control of $\Delta_t$ can provide a way to systematically explore the Hund's physics.

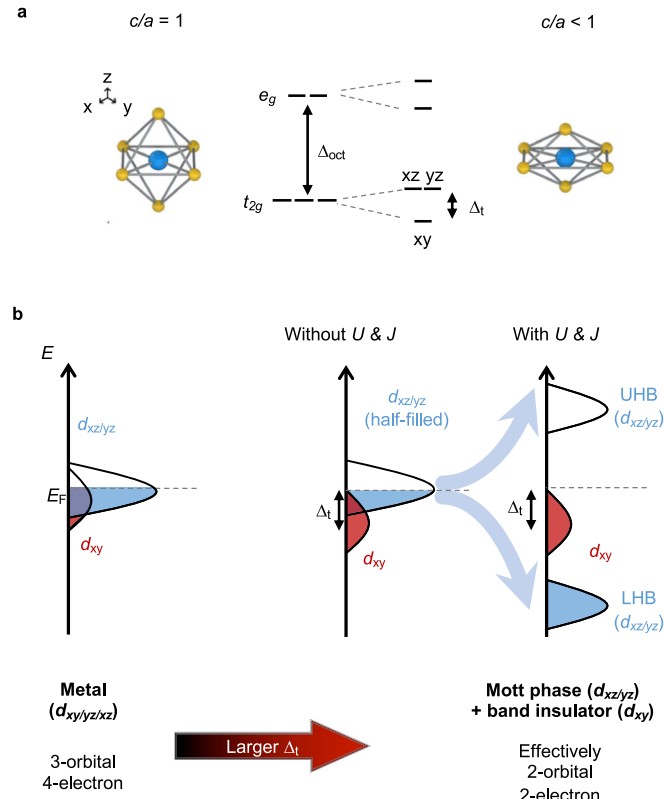

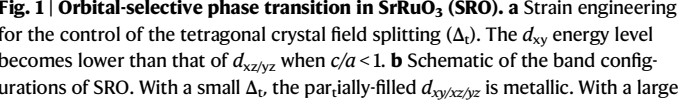

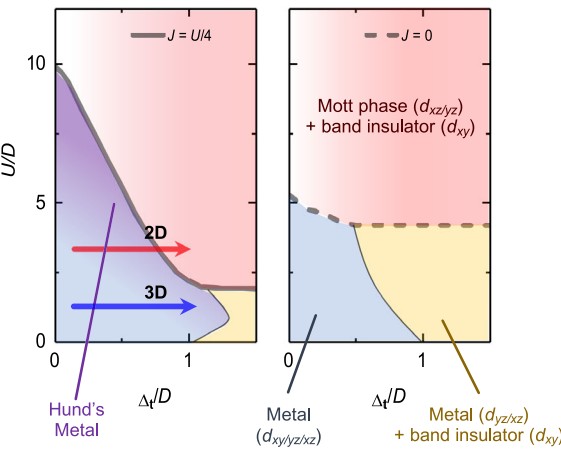

**Fig. 1 | Orbital-selective phase transition in SrRuO₃ (SRO). a** Strain engineering for the control of the tetragonal crystal field splitting ($\Delta_t$). The $d_{xy}$ energy level becomes lower than that of $d_{xz/yz}$ when $c/a < 1$. **b** Schematic of the band configurations of SRO. With a small $\Delta_t$, the partially-filled $d_{xy/xz/yz}$ is metallic. With a large $\Delta_t$, fully-filled $d_{xy}$ becomes band insulating, and half-filled $d_{xz/yz}$ becomes Mott insulating with sizable on-site Coulomb interaction ($U$) and Hund's rule coupling ($J$). **c** $U/D$ ($D$ is the half-bandwidth) and tetragonal crystal field splitting ($\Delta_t$)/$D$[14] phase diagrams for SRO with $J = U/4$ (left) and $J = 0$ (right).

## Results

### Symmetry-preserving strain engineering in 2D SRO

In this study, we observed an orbital-selective phase transition in an SRO monolayer with in-situ angle-resolved photoemission spectroscopy (ARPES). Thick SRO films (three-dimensional systems) did not exhibit metal-insulator transitions as shown in Supplementary Fig. 1. As stated earlier, the metal to band+Mott insulator transition is feasible with small control of $\Delta_t$ and large $U/D$ (red arrow in Fig. 1c). To study such effects more clearly, we investigated SRO monolayer systems, which contain a two-dimensional (2D) $RuO_2$ layer. When the thickness of the SRO film decreases and approaches the monolayer limit (2D), electron hopping along the z-direction diminishes[25–29]. The bandwidth of the $d_{xz/yz}$ orbitals can be further reduced by orbital-selective quantum confinement effects[30].

We used strain engineering to compress the oxygen octahedra along the out-of-plane direction in SRO monolayers. We used pulsed laser deposition to grow SRO layers on five different substrates, $(LaAlO_3)_{0.3}(Sr_2TaAlO_6)_{0.7}(001)$ [LSAT(001)], $SrTiO_3(001)$ [STO(001)], $Sr_2(Al,Ga)TaO_6(001)$ [SAGT(001)], $KTaO_3(001)$ [KTO(001)], and $PrScO_3(110)$ [PSO(110)]. They are known to have (pseudocubic) lattice constants of 3.868, 3.905, 3.931, 3.989, and 4.02 Å, respectively. Since the pseudocubic lattice constant of bulk SRO is 3.923 Å, the substrates impart −1.4%, −0.5%, +0.2%, +1.7%, and +2.5% epitaxial strain on the SRO monolayers. For convenience, we have indicated compressive (tensile) strain using a minus (plus) sign. One should note that these substrates have different oxygen octahedral rotation (OOR) patterns, so each SRO monolayer grown on the above-mentioned substrates could have a different OOR pattern[31–33]. Such undesirable occurrence of the complex structural modifications could hinder our systematic investigation of the orbital-selective phase transitions.

To suppress the structural complications of OOR, we developed a symmetry-preserving strain engineering technique. As shown in Fig. 2a, 10 unit cells (u.c.) of the $SrTiO_3$ (STO) layer were inserted between the SRO monolayer and substrates. This preserves the OOR pattern of the SRO monolayer while applying different epitaxial strains. Figure 2b, c show low-energy electron diffraction (LEED) results for the SRO monolayers under both compressive and tensile strains at 6 K. All samples showed LEED diffraction peaks at ($m + 0.5$, $n + 0.5$) ($m$, $n$: integers), indicating the existence of OOR along the out-of-plane axis. On the other hand, no sample showed ($m + 0.5$, $n$) and ($m$, $n + 0.5$) peaks, indicating that OOR did not occur along the in-plane axis. These LEED diffraction results indicate that all SRO monolayers on the substrates used exhibited OOR with $a^0a^0c^-$ crystal symmetry (Supplementary Fig. 2).

The atomic arrangements of the SRO monolayers were confirmed by scanning transmission electron microscopy (STEM). We covered the heterostructures with a 10-u.c. STO layer to protect the SRO layer from possible damage during STEM measurements. Figure 2d and Supplementary Fig. 3 show low-magnification STEM images in high-angle annular dark-field (HAADF) mode. High-magnification STEM images were acquired in the HAADF and annular bright-field (ABF) modes (Fig. 2e, f, respectively). The SRO monolayer did not show OOR along the in-plane axis, consistent with the LEED results. The analysis of lattice constants from the STEM results shows coherent strain state (Supplementary Fig. 4). Although most areas showed abrupt interfaces with the SRO single layer, we observed a few regions for which thickness inhomogeneities (i.e., 0 or 2 u.c. thickness of SRO) were observed (Supplementary Fig. 3). These inhomogeneous regions usually occurred near step terraces[34,35], which can break the continuity of the SRO monolayer for transport measurements. Instead, we used optical and in situ ARPES measurements to obtain reliable area-averaged responses. Possibilities

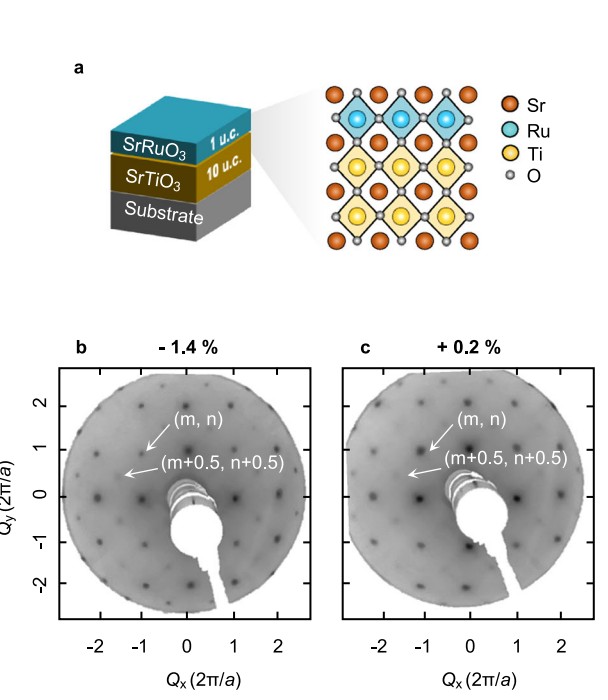

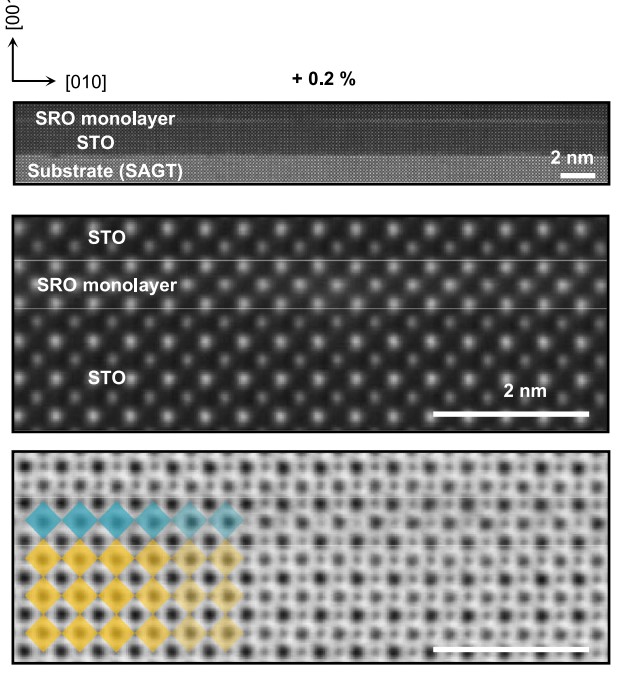

**Fig. 2 | Symmetry-preserved strained-SRO monolayers. a** Schematic diagram of an SRO monolayer on 10 unit cells (u.c.) of $SrTiO_3$ (STO) and substrate. The SRO monolayer has a single $RuO_2$ layer sandwiched between Sr−O layers. **b, c** Low-energy electron diffraction (LEED) images of SRO monolayers under −1.4 and +0.2% strain, with ($m$, $n$) and ($m + 0.5$, $n + 0.5$) peaks ($m$, $n$: integer). **d** Structural characterization of an SRO monolayer on an STO (10 u.c.)-SAGT substrate via high-

angle annular dark-field scanning transmission electron microscopy (HAADF-STEM). A 10 u.c. STO capping layer protects the SRO from damage during measurements. **e, f** HAADF (**e**) and annular bright-field (ABF) images (**f**). There is a single $RuO_2$ layer with abrupt interfaces without $RuO_6$ octahedron rotation along the in-plane axis.

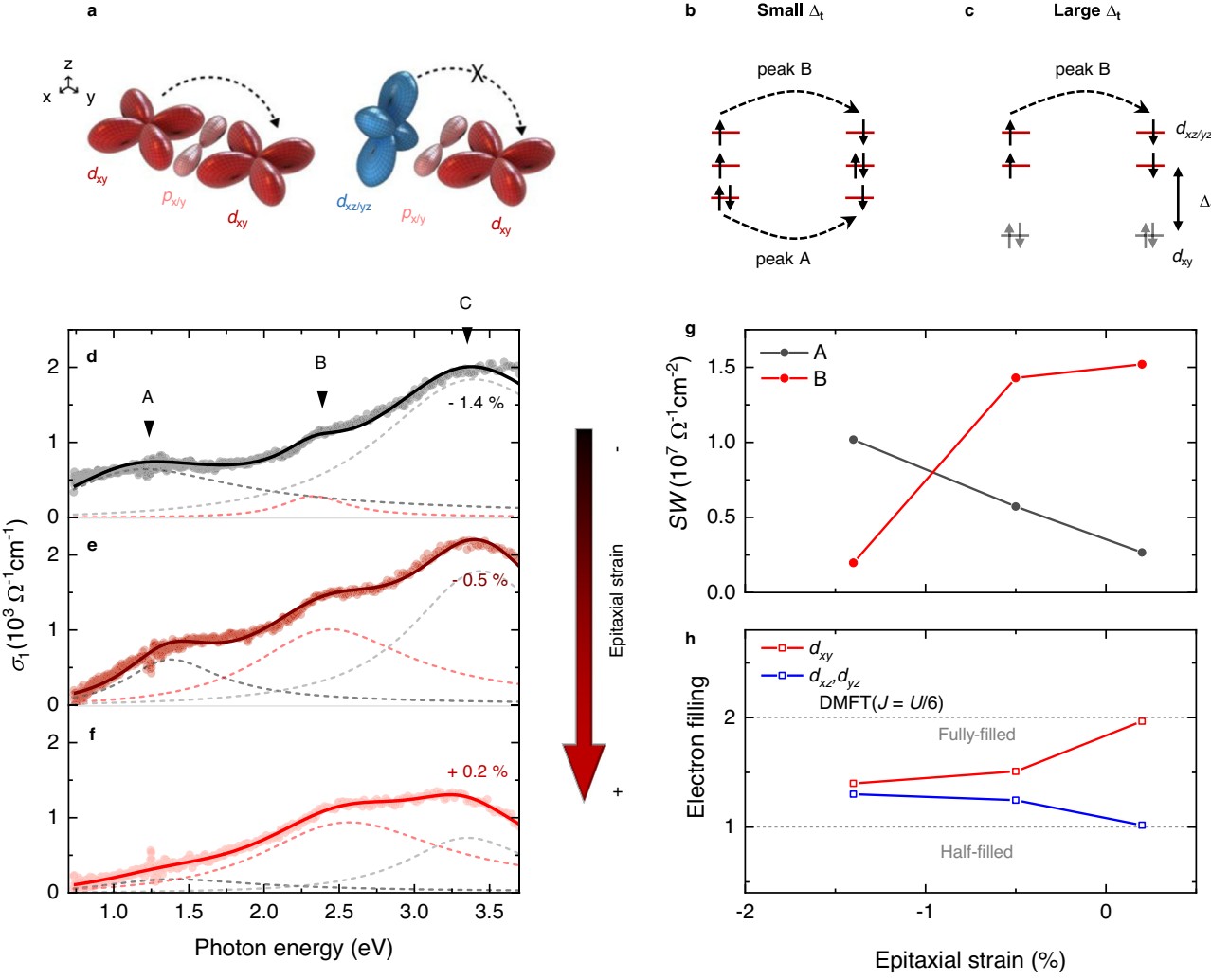

**Fig. 3 | Interatomic optical transitions and control of orbital occupancy.**
**a** Interatomic transitions between $d$-orbitals, for which the matrix elements can be quite large due to hybridization between Ru $d$ − O $p$ orbitals. Such transitions can occur only between the same orbitals. **b** Possible interatomic $d^4 + d^4 \rightarrow d^3 + d^5$ transitions in SRO with small $\Delta_t$. The energy costs for the two transitions differ because of $J$ (i.e., $U - J$ and $U + J$). **c** A possible interatomic $d^4 + d^4 \rightarrow d^3 + d^5$ transition in SRO with large $\Delta_t$. When $\Delta_t$ becomes large, $d_{xy}$ is fully filled while $d_{xz/yz}$ is half-filled. In

this case, only the $U + J$ transition can occur. **d–f** Optical spectra of SRO monolayers under various strains (−1.4, −0.5, and +0.2%) at 300 K. There are three peaks: A ($U − J$), B ($U + J$), and C (O $2p \rightarrow$ Ru $t_{2g}$). **g** Strain-dependent spectral weight ($SW$) of peak A and peak B. The $SW$ of peak A (B) decreases (increases) with an increase in strain to the plus side. **h** Strain-dependent electron filling calculated via dynamical mean-field theory (DMFT) with $J = U/6$ ($U = 2.7$ eV, $J = 0.45$ eV). When the strain reaches +0.2%, $d_{xy}$ becomes fully-filled, and $d_{xz/yz}$ becomes half-filled.

for contributions from the surface states of STO and SRO layers to the spectroscopic results are discussed in Supplementary Note 1.

## Orbital occupancy changes

To investigate the orbital occupancy changes, we explored the interatomic optical transitions between Ru⁴⁺ ions via ellipsometric spectroscopy. Upon absorption of a photon, an electron can hop from one Ru⁴⁺ ion to a nearest-neighbor ion ($d^4 + d^4 \rightarrow d^3 + d^5$ transition). The matrix element of such an interatomic transition can be significantly large due to hybridization between the Ru $d$ − O $p$ orbitals. Note that this interatomic transition can occur only between the same $t_{2g}$ orbitals due to the orbital geometry (Fig. 3a). Specifically, the transition from $d_{xz}$ to $d_{xy}$ (or $d_{yz}$) will be small, given that there is only a small overlap between the corresponding orbitals. Figure 3b, c display the allowed interatomic $d^4 + d^4 \rightarrow d^3 + d^5$ transition in SRO[36]. With a small $\Delta_t$, the three $t_{2g}$ orbitals are equally filled with four electrons. Then, interatomic transitions can occur at two photon energies ($U − J$ and $U + J$) (Supplementary Fig. 5)[36]. With a large $\Delta_t$, the $d_{xy}$ orbital becomes fully occupied, and the $U - J$ transition cannot occur.

We obtained an optical spectrum of the SRO monolayer under a strain of −1.4% (Fig. 3d). The spectrum exhibits three peaks (A–C). The A and B peaks are assigned to interatomic $d$-$d$ transitions with energy positions at $U − J$ and $U + J$, respectively. Peak C reflects a charge transfer transition from O $2p$ to Ru $t_{2g}$[37]. We fitted the spectrum (solid circles) with Lorentzian oscillators (dashed lines). The obtained peak positions were in good agreement with those previously reported for ruthenates[36,37]. The peak position difference between A and B is $2J$; $J$ was thus ~0.6 eV[36,38]. The extracted $U$ values from the optical results could be underestimated due to formation of excitons during optical processes[37]. The presence of both $U − J$ and $U + J$ peaks suggested that the compressively strained SRO monolayer contained partially-filled orbitals, which is similar to the situation for bulk SRO (Fig. 3b).

When the tensile strain is applied, a significant spectral weight ($SW$) change occurs, indicating strain-induced electron redistribution of the $t_{2g}$ orbitals (Fig. 3d–f). The change in $SW$ is plotted in Fig. 3g. As the tensile strain increases, the $SW$ of peak A decreases and that of peak B increases. For the tensile-strained SRO monolayer (+0.2%), peak A nearly disappears, suggesting that one of the bands (i.e., $d_{xy}$) becomes fully-filled. Dynamical mean-field theory (DMFT) calculations with

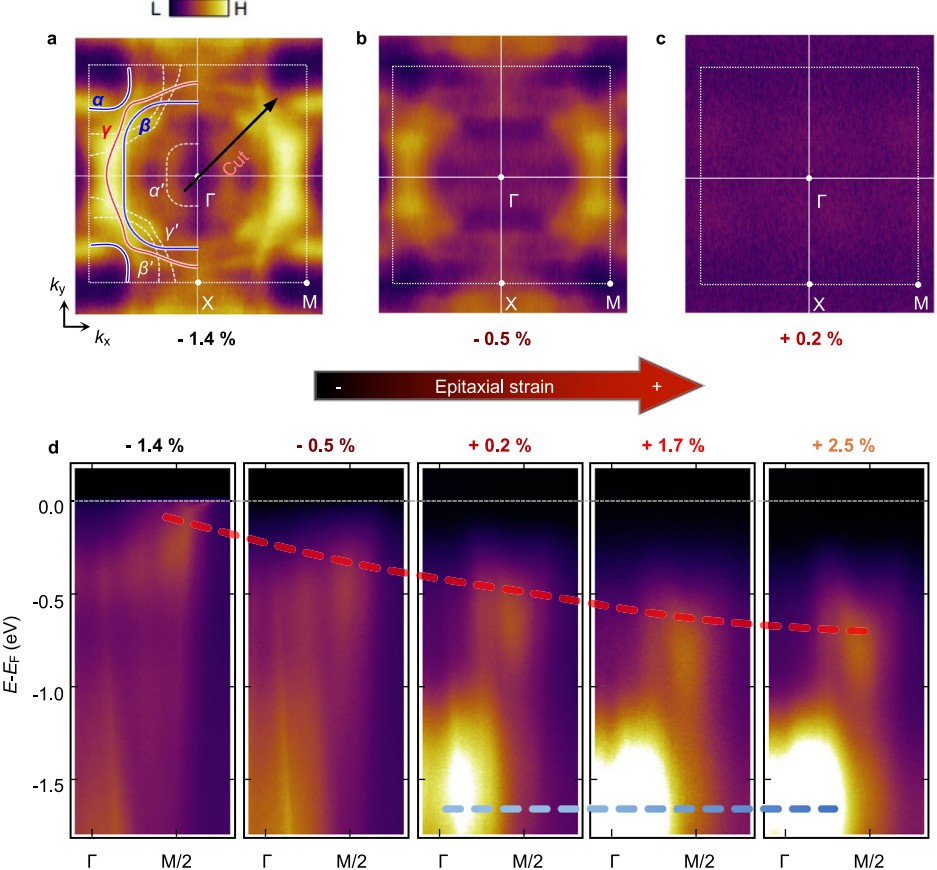

**Fig. 4 | Metal-insulator transition in SRO monolayers. a–c** Constant energy maps at the Fermi energy ($E_F$) of SRO monolayers under −1.4, −0.5, and +0.2% strain measured at 6 K. A metal-insulator transition (MIT) is as the strain changes. The notations ($\alpha$, $\beta$, and $\gamma$) for the three bands denoted by solid lines are adopted from the ones used for $Sr_2RuO_4$. Folded bands attributable to the rotation of $RuO_6$ are indicated by $\alpha'$, $\beta'$, and $\gamma'$ (dashed white lines). **d** $E$-$k$ data along the Γ−M line (denoted by the black arrow in (**a**) for −1.4, −0.5, +0.2, +1.7, and +2.5% strains.

$J = U/6$ ($U = 2.7$ eV, $J = 0.45$ eV) also reveals that the filling of $d_{xy}$ ($d_{xz}/d_{yz}$) orbital increases (decreases) with an increase in strain (Fig. 3h). When the strain is +0.2% or higher, $d_{xy}$ is fully filled while $d_{xz/yz}$ is half-filled (Supplementary Fig. 6). The control of orbital polarization in the Hund system can lead to an orbital-selective phase transition. Note that existence of such orbital polarization alludes to an insignificant role of the spin-orbit coupling in the orbital dependent Mott transition in the ruthenate films[39].

**Control of electronic structures with orbital differentiation**
The 2D metallic phase in the −1.4% strained SRO monolayer was studied by in-situ ARPES measurement. As SRO monolayers have 2D atomic arrangements (Fig. 2d–f), the electronic structure should be similar to that of $Sr_2RuO_4$ (a well-established quasi-2D system)[40–42]. Figure 4a shows the low-temperature constant energy map at $E_F$ for SRO monolayer with −1.4% strain. The experimentally obtained Fermi surface (FS) is similar to the schematic FS of $Sr_2RuO_4$ (shown as solid lines). Therefore, we hereafter follow the notation generally used for $Sr_2RuO_4$[40–42], in which the three bands at $E_F$ are labeled $\alpha$, $\beta$, and $\gamma$. The orbital characters of the $\alpha$ and $\beta$ bands are $d_{xz/yz}$, and that of $\gamma$ is $d_{xy}$.

Energy maps of the SRO monolayers show the strain-induced MIT. We measured constant energy maps for SRO monolayers at $E_F$ under epitaxial strains of −1.4, −0.5, and +0.2%. Compressively strained SRO monolayers (−1.4 and −0.5%) exhibit a nonzero density of states (DOS) at $E_F$, indicating that they are metallic (Fig. 4a, b). In contrast, tensile-strained SRO monolayers (+0.2%) have nearly zero DOS at $E_F$, showing that they are in an insulating state (Fig. 4c). These strain-dependent

DOSs at $E_F$ in the SRO monolayer indicate that MIT occurs at a strain between −0.5% and +0.2%, which agrees well with the optical spectra of the SRO monolayer (Fig. 3g, h).

To elucidate the mechanism of the MIT, we focused on strain-dependent $E$-$k$ dispersion along the Γ−M line indicated by the black arrow in Fig. 4a (Fig. 4d). Two major features with epitaxial strain dependence (−1.4, −0.5, +0.2, +1.7, and +2.5%) are observed. First, the energy level of the band near M/2 (red dashed line) moves smoothly to higher binding energies with increasing tensile strain. Second, the spectral weight ($SW$) near Γ (blue dashed line) at $E = E_F − 1.6$ eV appears abruptly at a strain of +0.2% and is enhanced above +0.2% tensile strain.

Orbital characters of the bands in SRO monolayers were identified by examining constant energy maps below $E_F$ for SRO films under a +0.2% strain. Figure 5a shows an energy distribution curve (EDC) from the Γ−M line of the SRO monolayer (+0.2%). The constant-energy maps at $E−E_F = −0.5$ eV (red dashed line) and −1.6 eV (blue dashed line) were acquired for two experimental geometries, as shown in Fig. 5b. Note that the photoemission intensity for the $d_{xy}$ orbital should vary with respect to the azimuthal angle $\varphi$, whereas that for the $d_{xz/yz}$ orbital should change little with respect to $\varphi$ (Supplementary Note 2). At $E_F − 0.5$ eV, we observe a significant difference in the photoemission intensity between $\varphi = 0°$ and 45°. In particular, the intensity along the Γ−M line with $\varphi = 45°$ (Fig. 5d) was much lower than that with $\varphi = 0°$ (Fig. 5c). On the other hand, at $E_F − 1.6$ eV, the photoemission intensity at $\varphi = 0°$ and 45° show little difference (Fig. 5e, f). These observations can be understood by considering the matrix element[43]. Therefore, we conclude that the orbital characteristics of near $E_F − 0.5$ eV band is

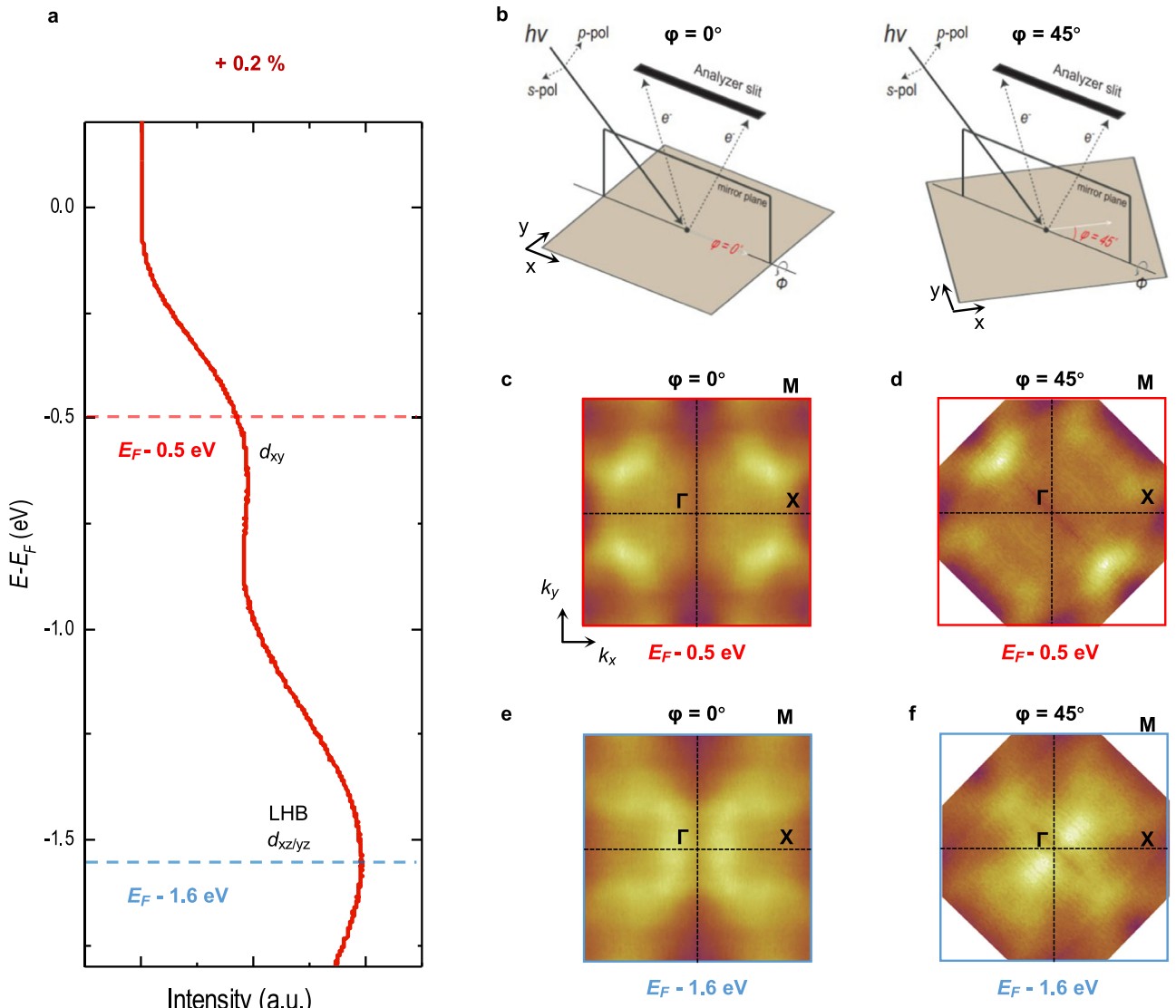

**Fig. 5 | Orbital contributions of the bands below the Fermi energy ($E_F$) in an SRO monolayer under + 0.2% strain. a** Energy distribution curve of the SRO monolayer (+0.2%) from the Γ−M line. To investigate the orbital character, we measured the constant energy maps at two azimuthal angles ($\varphi$). The maps were obtained at $E_F − 0.5$ eV (red dashed line) and $E_F − 1.6$ eV (blue dashed line) using dominantly $s$ polarized light. **b** Experimental geometry for $\varphi = 0°$ and 45°. **c, d** Constant energy maps at $E_F − 0.5$ eV with $\varphi = 0°$ (**c**) and 45° (**d**). **e, f** Constant energy maps at $E_F − 1.6$ eV with $\varphi = 0°$ (**e**) and 45° (**f**).

dominantly of $d_{xy}$ while that for the band near $E_F − 1.6$ eV is dominantly of $d_{xz/yz}$.

This orbital differentiation can be used to explain the band structure evolution in Fig. 4d, which confirms the rationale illustrated in Fig. 1b. As the tensile strain is applied, the $d_{xy}$ band near M/2 moves steadily toward the high binding energy side and becomes a band insulator at the MIT (red dashed line). We also observe that near Γ ($d_{xz/yz}$) spectral weight suddenly appears close to the MIT (blue dashed line). As $d_{xz/yz}$ becomes half-filled with tensile strain (Fig. 3h), the band near Γ should be the lower Hubbard band (LHB). Therefore, the MIT occurrs with orbital differentiation. The two gaps open nearly simultaneously at the MIT: a band insulating gap for $d_{xy}$ and a Mott gap for $d_{xz/yz}$[38,44]. More details on the energy positions of LHB and UHB are discussed in Supplementary Note 3.

Careful analysis of the ARPES spectra also reveals how $\Delta_t$ can indeed be tuned via our symmetry-preserving strain engineering technique. EDCs of the Γ−M cut for different strain values are shown in Supplementary Fig. 7. Peaks are seen at energies from $E_F − 0.8$ to $− 0.4$ eV (from $d_{xy}$) and $− 1.6$ eV (from the LHBs of $d_{xz/yz}$). The peak position of the $d_{xy}$ band ($\zeta$) was obtained via

Gaussian fitting. The $\zeta$ value (black circles) increases with the tensile strain, consistent with the density functional theory (DFT) (pink squares). Since the LHB energy varies little with the strain, the change in the $\zeta$ value should be from the change in $\Delta_t$. Therefore, the systematic $\zeta$ change in the ARPES indicates that $\Delta_t$ increases with an increase in tensile strain, as expected from Fig. 1b.

We also tried to change the orbital occupancy by dosing potassium (K) on the SRO monolayer surface. K atoms donate electrons without significantly perturbing the SRO monolayer structure. The coverage of the K layer was controlled from 0.0 monolayer (ML) to 1.0 ML. Supplementary Fig. 8 shows K-coverage-dependent ARPES data for the SRO monolayer in a band ($d_{xy}$) +Mott($d_{xz/yz}$) insulating state with +2.5% strain. With electron doping, the DOS for the LHBs ($d_{xz/yz}$) decreases while the DOS near $E_F$ increases. This indicates that the correlation-induced Mott gap collapses as the system moves away from the half-filled $d_{xz/yz}$. However, the SRO monolayer with 1.0 ML K-coverage is still insulating (almost zero DOS at $E_F$), possibly due to strong Anderson localization effects in the 2D system[45].

## Discussion

In summary, we demonstrated tuning of the crystal field of a SRO monolayer with symmetry-preserving strain engineering. Given the nature of 2D materials, the modulated $\Delta_t$ induces compulsory changes in the orbital occupancy, which results in a dramatic orbital-selective phase transition. Remarkably, a simultaneous MIT with orbital differentiation was observed via in situ ARPES measurements, in which one of the bands becomes band insulating while the other opens a Mott gap. Note that $J$ and crystal field energy scales compete each other in terms of the orbital polarization. Therefore, the observed orbital-selective phase transition induced by the crystal field tuning in Hund's system will provide an insight on the role of $J$.

Our symmetry-preserving strain engineering technique can be applied to studies of various multi-orbital physics in numerous transition metal oxide monolayers[46,47]. According to recent theoretical and experimental studies[3–12], Hund's physics is crucial in the description of novel quantum phenomena such as unconventional superconductivity and magnetism as well as potential device applications. However, impurity problems and complex structural distortions in materials have hindered manipulation and observation of Hund's physics, such as orbital-selective Mott phases. In this context, our strategy can be used extensively for precise control of materials with simplified structures and negligible inhomogeneity problems. Thus, our work provides a way to investigate novel phenomena in multi-orbital systems and promote multi-orbital-based device applications.

## Methods

### Sample preparation

STO and SRO epitaxial layers were grown on various substrates via PLD. The targets were single-crystalline STO and polycrystalline SRO. A KrF excimer laser ($\lambda = 248$ nm; coherent) was operated with a repetition frequency of 2 Hz. For deposition of the STO and SRO layers, the laser energies were 1.0 and 2.0 J cm$^{-2}$, respectively. The deposition temperature was 700 °C, and the oxygen pressure was 100 mTorr. The film thickness was controlled at the atomic scale by monitoring reflection high-energy electron diffraction patterns (Supplementary Fig. 9).

### Scanning transmission electron microscopy

An electron-transparent STEM specimen was prepared via focused ion-beam milling (Helios 650 FIB; FEI) and further thinned via focused Ar-ion milling (NanoMill 1040; Fischione). Cross-sectional STEM images were acquired at room temperature using an instrument corrected for spherical aberration (Themis Z; Thermo Fisher Scientific Inc.) and equipped with a high-brightness Schottky-field emission gun (operating at an electron acceleration voltage of 300 kV). The semiconvergence angle of the electron probe was 17.9 mrad. The collection semiangle for ABF was 10–21 mrad.

### In situ angle-resolved photoemission spectroscopy

ARPES measurements were performed using home-built laboratory equipment comprising an analyzer (Scienta DA30) and discharge lamp (Fermion instrument). He-I$\alpha$ ($h\nu = 21.2$ eV) light partially polarized in a linear vertical direction (70%) was used. As all of the substrates used in this study were insulators, ARPES measurements could be hampered by charging effects arising from the sample geometry (Fig. 2a). Thus, we inserted a conducting layer of 4-u.c.-thick SRO between the substrate and 10 u.c. STO layer[25]. The heterostructures are fully strained to substrates as shown in Supplementary Fig. 10. The STO layer decoupled the electronic structure of the topmost SRO monolayer from that of the inserted SRO conducting layer[25]. After growth, the SRO heterostructures were transferred to an ultrahigh vacuum (UHV; <1.0 × 10$^{-10}$ Torr) environment without air exposure and annealed at 570 °C for 20 min immediately before measuring ARPES. This annealing process makes the surface quality of the sample suitable for ARPES

measurements (Supplementary Figs. 11, 15). It was previously shown that oxygen vacancy effects are negligible for SRO films annealed under the optimized condition[25].

### Low-energy electron diffraction

All LEED data were collected using a SPECS ErLEED 1000-A. The base pressure of the UHV chamber was maintained below 8.0 × 10$^{-11}$ Torr. To match the results of the ARPES and LEED measurements, sample preparation before LEED measurements was identical to that before ARPES (preannealing of 570 °C for 20 min at <1. 0 × 10$^{-10}$ Torr). The UHV chamber for LEED measurements was connected to the chambers used for ARPES measurements and sample growth; the sample was not exposed to air before the LEED measurements.

### Optical spectroscopy

Optical spectroscopic characterization was performed using a spectroscopic ellipsometer (M-2000 DI; J.A. Woollam Co.). The reflectance of the bare substrate, STO (10 u.c.)/substrate, and SRO (1 u.c.)/STO (10 u.c.)/substrate were measured separately, and the optical conductivity was extracted for each layer (Supplementary Fig. 12). It was challenging to obtain the optical spectra of samples under higher tensile strain (+1.7 and +2.5%). The as-received KTO(001) substrate had 3–4 nm-deep surface holes. Although ARPES measurements were not seriously affected by such holes, obtaining reliable optical data from spectroscopic ellipsometry was challenging. In addition, as PSO(110) (+2.5%) single crystalline substrates have orthorhombic structures, it was quite difficult to subtract the strongly anisotropic responses.

### Density functional theory and dynamic mean-field theory calculations

We carried out density functional theory (DFT) calculations within the Perdew–Burke–Ernzerhof exchange-correlation functional revised for solids using VASP code[48]. For the lattice constant, the experimental value of the corresponding substrate material was used for each strain configuration. We used a 600 eV plane-wave cutoff energy and 6 × 6 × 1 k-points for all DFT calculations. The internal atomic positions of the three layers closest to the surface were fully relaxed until the maximum force was below 5 meVÅ$^{-1}$. Maximally localized Wannier functions[49] for $t_{2g}$ bands were constructed for the tight-binding model to be used in DMFT calculations. We performed a single-site DMFT calculation on top of the Wannier Hamiltonian with a continuous-time QMC hybridization-expansion solver implemented in TRIQS/CTHYB. Supplementary Fig. 13, Fig. 14 show the detailed band structure obtained from the Wannierization for the $t_{2g}$ orbitals and the electron hopping term of $d_{xy}$-$d_{xy}$ and $d_{xz}$-$d_{xz}$ (or $d_{yz}$-$d_{yz}$).

## Data availability

Source data are provided with this paper. Other data is available from the authors upon request. Source data are provided with this paper.

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

## Acknowledgements

All authors are grateful for the major support provided by the Research Center Program of the Institute for Basic Science of Korea (grant no. IBS-R009-D1). S.H. and C.K. acknowledge the supported by the National Research Foundation of Korea (NRF) grant funded by the Korea government (MSIT) (No. 2022R1A3B1077234). C. Sohn was supported by a National Research Foundation of Korea (NRF) grant funded by the Korean government (MSIT) (Creative Materials Discovery Program-No.2017M3D1A1040834) and the Ministry of Science and ICT (2020R1C1C1008734). S.-S.B.L. was supported by the National Foundation of Korea (NRF) grant funded by the Korea government (MSIT) (No. 2022R1F1A107452211). S. Lee and M. Kim acknowledge financial support from the Korean government through the National Research

Foundation (grant no. 2017R1A2B3011629). Cs-corrected STEM was performed at the Research Institute of Advanced Materials (RIAM) of Seoul National University.

## Author contributions

E.K.K. and T.W.N. conceived and designed the project. E.K.K. fabricated and characterized the films via optical spectroscopy. S.H. characterized the films via ARPES and LEED. C.S., J. Son., J. Song, and T.W.N. contributed to the optical spectroscopy data analysis. S.H., D.K., and Y.K. contributed to the ARPES measurements. S.L. and M.K. performed STEM and analyzed the data. S.H., B.S., J.R.K., and C.K. contributed to the analysis of the ARPES results. C.H.K. S.S.B.L. contributed to the DFT and DMFT calculations. E.K.K., S.H., C.H.K., C.K., and T.W.N. wrote the paper with contributions and feedback from all authors. T.W.N. initiated the study and was responsible for the overall research direction. E.K.K. and S.H. contributed equally to this work.

## Competing interests

The authors declare no competing interests.
