## [Peer Review File · Nature Communications]

REVIEWER COMMENTS

Reviewer #1 (Remarks to the Author):

The Manuscript by Ko et al discusses the manipulation of orbital occupation in a ruthenate using epitaxial strain. The authors controlled the relative occupancy of the three t_{2g} orbitals by controlling epitaxial strain. They show using a combination of photo-emission and optical experiments that a monolayer ruthenate film undergoes a MIT as the strain is modulated from being compressive to tensile. In general, the work is well done and I think that this will be a good addition to the literature on orbital-selective phase transitions. However, I suggest that the following be addressed before publication:

i) The correlation between strain and spectral changes are based on theoretical strain values. I did not see any structural data that measures the actual strain in the SRO monolayers. Given that there is an SRO layer and an STO layer on substrate - the assumption of coherent strain transmission needs to be supported.

ii) The role of surface changes and influence of surface states was not discussed at all - for a 1 ML film, accounting for contributions from the surface may be necessary. At the least a discussion on possible contributions from the surface and how that would influence the conclusions of this work would be useful.

iii) the films were annealed at 570C for 20 mins in the ARPES chamber before measurements - oxygen vacancy creation is a possibility under these conditions. How did the authors rule out contributions from oxygen vacancies?

iv) There is a lot of published work on orbitally controlled physics in oxides: For example, <https://www.science.org/doi/10.1126/science.1149338>; <https://journals.aps.org/prl/abstract/10.1103/PhysRevLett.94.156402>; <https://www.nature.com/articles/nphys2733>. I feel that the authors; <https://www.nature.com/articles/s41467-019-08472-y>. I suggest that this be captured - The introduction sounds like strain-controlled crystal field splitting and the associated orbital occupancy changes is a new approach. While this is a still a useful addition, a discussion about some of the previous approaches including charge transfer and strain could be useful.

v) A few minor suggestions:

1) Line 59 - 61: t_{2g} orbital degeneracy is lifted for both compression and elongation - not just elongation.

2) Line 95: '...strain engineering to elongate oxygen octahedra..' - along which direction?

3) Line 137-139, Isn't the statement: "the transition from dx_z to dx_y (or dy_z) not allowed" only true in the absence of any p-d hybridization?

4) Lines 177-179: "The MIT occurs at a strain of approximately +0.2%" I think that the data does not support this statement. The MIT could be anywhere between -0.5% and +0.2%.

Reviewer #2 (Remarks to the Author):

The authors of this manuscript report on a metal insulator transition driven by orbital selective occupancy of orbitals in strained monolayer SrRuO₃ films. The transition is controlled by the degree of tetragonal distortion of the unit cell imposed by growing the monolayers on top of a variety of substrates that span a range between compressive to tensile stress. Based on ellipsometric and ARPES experiments, the authors infer a Mott transition taking place in the dx_z/dy_z subset as the tensile strain grows sufficiently. The latter is ruled by strong correlations dictated by the ratio between Hund's coupling J and the U correlation energy.

To obtain reliable results, the authors devise a way to control the field splitting due to the tetragonal distortion, by developing a symmetry-preserving strain engineering technique (i.e., by inserting a STO layer between substrate and SrRuO₃ monolayer). The success of this strategy was confirmed by LEED diffraction patterns and transmission electron microscopy. This guaranteed that the relevant parameter was the c/a distortion and make reliable the comparison between the samples with different strain state.

They subsequently used ellipsometry over a wide range of frequencies and identified three main peaks. The authors propose specific transitions between electronic states in the t_{2g} manifold, ruled by parameters U and J . They assumed that electrons are transferred to neighboring sites in the lattice by interacting with light, which I think is a reasonable and justified hypothesis. Based on the

evolution of the spectral weight with strain, they interpreted the results as coming from the relative shift in energy of d_{xy} and d_{xz}/d_{yz} orbitals induced by tetragonal distortions and electronic correlations.

The authors used data from ARPES to determine the orbital character of states below the Fermi energy. They checked that orbitals closer to EF are d_{xy} , while those down in energy are of d_{xz}/d_{yz} character.

I think that this is an interesting work that combines efforts of materials science with fundamental insights into the physics of correlated systems. I have a couple of comments:

First, I suggest making a bit clearer the discussion about the orbital hierarchy inferred from ARPES. The experiments show that under positive strain, where $c/a < 1$, the d_{xz}/d_{yz} orbitals are further away from EF than d_{xy} orbitals. This is not compatible with an orbital hierarchy fully determined from c/a tetragonality, since for $c/a < 1$ one should expect d_{xy} states to be lower in energy (as depicted in Fig. 1b). This observation could be used by the authors to reinforce their viewpoint. A Mott transition with UHB/LHB splitting in the d_{xy}/d_{xz} sector would be compatible with ARPES observations, which otherwise could not explain by pure distortion effects. On the other, I suggest the authors to discuss the c/a parameter as extracted from STEM and include this discussion in the manuscript.

My second comment is about the role of spin-orbit coupling (SOC), which is relevant in 4d ions (SOC is in the range of 0.2-0.3 eV for these ions) but is not discussed at all in the manuscript. The physics of SOC in t_{2g} systems is described in some references (see, e.g., Streltsov et al., Phys. Rev X 10, 031043 (2020) or Khomskii and Streltsov, Chem. Rev. 121, 5, 2992 (2021)). The Δ_t “crystal field” parameter, originated by the c/a distortion, is in the range of a few hundreds of meV (as inferred from EDCs data in Fig. S5). This is not far from energy scales of SOC in 4d or from exchange coupling energy. In particular, my question is whether the LHB band (inferred from ARPES experiments, see Figures 5 or S5) is split because of SOC. Maybe the energy resolution is not enough to discriminate the splitting, but in any case, I suggest the authors to discuss the role of spin-orbit coupling in the physics of the orbital-selective phase transitions discussed in this manuscript.

Reviewer #3 (Remarks to the Author):

The authors design and realize a smart way to tune the band structure of SrRuO₃ by growing thin films on top of different substrates avoiding distortions in the oxygen octahedra. This allows them to detect a metal insulator transition as a function of the strain induced by the substrate. The authors

adscribe the transition to an orbital selective Mott transition due to Hund's physics. I have some concerns with several of their arguments and discussion:

- The phase diagram in Fig. 1^a is confusing, and it seems to me partially wrong. As far as I understand the $J/4$ and $J=0$ have to be compared with the negative crystal field splitting in Ref. 14 (figs. 1a and 3a). There are however several differences: (i) Why is the crystal field here given in eV? Ref.14 seems to be given in units of the half-bandwidth D . (ii) The $J=0$ boundaries do not show the same tendencies as in Fig.3a of Ref.14. (iii) Moreover plotting the $J=U/4$ and $J=0$ is confusing especially because they color the phase diagram following the $J/4$ lines. I suggest the authors plot the two values of Hund's coupling separately and be careful when plotting the lines.

- I think that the experimental MIT is robust. As far as I understand the authors claim that this MIT corresponds to the transition between a metal with both d_{xy} and $d_{zx/dyz}$ orbitals partially filled and an insulator with d_{xy} fully occupied and $d_{zx/dyz}$ half-filled. They argue that this is a proof of Hund's physics. While Hund's coupling increases the range of U/W in which this transition can happen, as observed in Fig.3a of Ref.14 this transition can also happen at $J=0$. Therefore in my opinion the authors cannot claim that they observe Hund's physics.

- I think that the numbers of the interactions do not match well the arguments from the authors. If I understand the arguments by the authors, from Fig.3 they obtain $U-J \sim 1.2$ eV, $U+J \sim 2.4$ eV, thus $J \sim 0.6$ eV and $U \sim 1.8$ eV. But in Fig.5 the lower Hubbard band is centered around -1.6 eV. This value can be expected to be around $U/2$. Then I would expect U to be around 3.6 eV, much larger than the value obtained in Fig.3. Am I wrong with the interpretation of the results? Could the authors comment on this? - Can the authors give more detailed information about the effective tight-binding that they obtain from the Wannierization for the t_{2g} orbitals?. In particular how much does it change the crystal field with the strain? Is there any change in the bandwidth of the non interacting band? And particularly for the d_{zx} and d_{yz} subspace? Also, which value of U do the authors use in the calculations? How large is the orbital dependent bandwidth? I think that I have not seen any of this information either in the main text or in the supplemental material.

- Can the authors give in Fig. 3h the DMFT electron filling for the cases with strain 1.7 and 2.5%. This is relevant for the interpretation of the ARPES results.

Due to all these concerns, in my opinion the manuscript, at least in its present version is not suitable for publication in Nature Comm.

Dear Reviewers

We greatly appreciate insightful and helpful comments from all the reviewers. Based on the suggestions, we performed additional experiments as well as theoretical calculation. The new results were added as supplementary figures. Also, we have gone through sincere efforts to revise the manuscript to prevent grammatical errors, making the manuscript more readable and explicit. We believe that, thanks to the comments, our manuscript is much improved and ready for resubmission.

Please find below our responses to the reviewers' comments. In **Part A**, we have our point-by-point responses to the reviewers' comments. In **Part B**, a list of changes in the revised manuscript and supplementary information are provided.

Part A: Responses to the comments

Reviewer #1

The Manuscript by Ko et al discusses the manipulation of orbital occupation in a ruthenate using epitaxial strain. The authors controlled the relative occupancy of the three t_{2g} orbitals by controlling epitaxial strain. They show using a combination of photo-emission and optical experiments that a monolayer ruthenate film undergoes a MIT as the strain is modulated from being compressive to tensile. In general, the work is well done and I think that this will be a good addition to the literature on orbital-selective phase transitions. However, I suggest that the following be addressed before publication:

We thank the reviewer for the careful review of our manuscript and also for acknowledging the importance of our work. We have answered all questions from the reviewer as shown below.

1. The correlation between strain and spectral changes are based on theoretical strain values. I did not see any structural data that measures the actual strain in the SRO monolayers. Given that there is an SRO layer and an STO layer on substrate – the assumption of coherent strain transmission needs to be supported.

In the original manuscript, we *assumed*, without providing experimental structural information, that both SrRuO₃ (SRO) and SrTiO₃ (STO) layers on substrates were fully strained. We agree that providing structural data regarding the coherent strain is essential. To measure the in-plane lattice constants of films, we performed reciprocal space mapping (RSM) experiments with X-ray diffraction (XRD). We also obtained the lattice constants from scanning transmission electron microscope (STEM) data.

Figure A1 shows RSM results of STO(10 u.c.)/SRO(1 u.c.)/STO(10 u.c.)/SRO(4 u.c.)/STO(10 u.c.) on LSAT(001) and KTO(001) substrates. Here, LSAT(001) and KTO(001) give -1.4% and +1.7% of strain compared to the bulk SRO lattice constant, respectively. Note that we used SRO (1 u.c.)/STO(10 u.c.)/SRO(4 u.c.)/STO(10 u.c.)/substrate heterostructures for angle-resolved photoemission spectroscopy (ARPES) to avoid the charging problem during the measurements. We measured the (103) diffraction peak of the substrate and STO layers. Identical q_x values for the substrate and STO layer indicate that the films are fully strained to the substrate for both compressive and tensile strains. While SRO layers were too thin to observe diffraction peaks, it is reasonable to assume that SRO layers are also fully strained to the substrate as both the STO buffer and capping layers are fully strained to the substrate.

Figure A1 XRD scans for STO (10 u.c.)/SRO (1 u.c.)/STO (10 u.c.)/SRO (4 u.c.)/STO (10 u.c.) on LSAT(001) and KTO(001) substrates. RSM scans of the heterostructures on LSAT(001) (a) and KTO(001) (b), which give -1.4 % and +1.7% strain, respectively. The red dashed lines indicate the q_x values of the (103) peak of the substrates. The in-plane lattice constants of STO layers match those of the substrates, indicating films are fully strained to the substrates.

Furthermore, we estimated the lattice constants from STEM results measured in the high-angle annular dark field (HAADF) mode. **Figure A2** shows the lattice constants for the films on LSAT(001) and SAGT(001) substrates. They give -1.4% and +0.2% strain compared to the bulk SRO lattice constant, respectively. We plotted a/a_{sub} values in **Figures A2c** and **A2d** as a function of the atomic layer number, where a_{sub} is the in-plane lattice constant of the substrate. The in-plane lattice constants of both SRO and STO layers are fully strained to the substrates, which is consistent with the RSM results.

Figure A2 Lattice constant analysis from STEM results for STO(10 u.c.)/SRO(1 u.c.)/STO(10 u.c.) on LSAT(001) and SAGT(001). **a,b**, STEM-HAADF images of STO(10 u.c.)/SRO(1 u.c.)/STO(10 u.c.) on LSAT(001) (**a**) and SAGT(001) (**b**). **c,d**, In-plane lattice constant as a function of the atomic row number. Both SRO and STO layers are fully strained to the substrate.

We made the following changes to the revised manuscript.

(New figures) Figures A1 and A2 as Supplementary Fig. 10, and Fig. 4.

(Added, in the 9th paragraph) The analysis of lattice constants from the STEM results shows coherent strain state (**Supplementary Fig. 4**).

(Added, in Methods section) The heterostructures are fully strained to the substrates as shown in **Supplementary Fig. 10**.

2. The role of surface changes and influence of surface states was not discussed at all – for a 1 ML film, accounting for contributions from the surface may be necessary. At the least a discussion on possible contributions from the surface and how that would influence the conclusions of this work would be useful.

The reviewer raised an interesting issue of surface state/condition effect. First of all, we would like to point out that SRO monolayers are grown on STO(10 u.c.) for all experiments. It is known that oxygen deficiency on the surface results in metallic surface states for STO [S. Soltani *et al.*, *Physical Review B* **95**, 125103 (2017), S. M. Walker *et al.*, *Physical Review Letter* **113**, 177601 (2014)]. However, it was previously shown that the surface of STO(10 u.c.) is very insulating without surface states [Sohn *et al.*, *Nature Communications* **12**, 6171 (2021)], meaning that the amount of oxygen deficiency, if any, is very small. Therefore, we address influence of the surface condition to spectroscopic results of SRO monolayer in terms of thickness inhomogeneity, surface termination, and possible surface chemical adsorbates.

To investigate the thickness inhomogeneity of our SRO monolayers, we performed both macroscopic (atomic force microscopy (AFM) measurements and reflection high-energy electron diffraction (RHEED)) and microscopic measurements (STEM). **Figure A3** is an AFM image of SRO(1 u.c.)/STO(10 u.c.) on STO(001) substrate, which shows step terraces. **Figure A4** shows RHEED patterns of substrates and SRO monolayers. Considering our experimental geometry (electron-beam energy of ~ 15 keV and incidence angle of $\sim 3^\circ$ from the surface), the probing depth should be around 0.4 nm, which is the thickness of one layer of SRO [D. Tang *et al.*, *Physical Review B* **50**, 24 (1994)]. RHEED results of SRO monolayers on various substrates always show 2D-like patterns, indicating that the surface of SRO monolayers has very small roughness (i.e. small thickness inhomogeneity). STEM results also show that a film is mostly 1 u.c. thick. However, some regions in STEM results show thickness inhomogeneities (i.e., 0 or 2 u.c. of SRO) (**Figure A5, Supplementary Fig. 3**). Such inhomogeneities can significantly affect some experimental measurements such as transport properties. On the other hand, area-averaged signal is obtained in ARPES experiments. As our films are mostly 1 u.c. thick, we believe the average spectroscopic response should represent that of an SRO monolayer.

Figure A3 AFM image of SRO (1 u.c.)/STO(10 u.c.) on STO(001) substrate.

Figure A4 RHEED patterns along [100] direction of substrates and films on substrates. The first row shows RHEED image from substrates while the second row is RHEED images of SRO (1 u.c.)/STO (10 u.c.)/SRO (4 u.c.)/STO (10u.c.) on substrates.

Figure A5 Scanning transmission electron microscopy (STEM) images of SRO on LSAT(001). **a**, Structural characterization of an SRO monolayer on a STO (10 u.c.)-LSAT substrate via STEM in the high-angle annular dark-field (HAADF) mode. A 10 u.c. STO cap protects the SRO from damage during measurements. **b**, HAADF and **c**, annular bright-field images (**c**); A single RuO_2 layer with abrupt interfaces is revealed, without RuO_6 octahedron tilting. **d**, HAADF-STEM images of different regions in SRO on LSAT(001). Most of the regions have atomically sharp interfaces with an SRO monolayer, as shown in **Fig. 2** (main text). However, a few regions exhibit thickness inhomogeneities. The red arrows indicate regions with discontinuous RuO_2 layers (0 u.c. SRO). The blue arrows indicate regions with RuO_2 double layers (2 u.c. SRO).

Second, the electronic structure can differ depending on the surface termination. For example, the oxygen octahedral symmetry may be broken at the surface if the film is B-site terminated [H. G. Lee et al., *Advanced Materials* **32** 1905815 (2020)]. However, due to the highly volatile Ru_xO_y at high temperature, SRO films grown by PLD are always Sr-terminated (A-site terminated) regardless of substrate termination [G. Koster et al., *Reviews of Modern Physics* **84**

(2012)]. Therefore, electronic structure variation due to different terminations can be ruled out for SRO films.

Finally, to minimize surface chemical adsorbates, we post-annealed as-grown SRO samples without exposing them to the air before ARPES measurements. Such process allows us to have a high quasi-particle peak to high-binding peak intensity ratio (I_{QP}/I_{HB}) as reported in B. Sohn *et al.*, Nature Communications **12**, 6171 (2021). Moreover, we performed LEED experiments before/after the post-annealing process, with the results shown in **Figure A6**. We can see a clear enhancement in sharpness of LEED peak after the post-annealing. From this, it was confirmed that annealing was helpful in improving the surface quality of samples.

Figure A6. Surface structure characterization of monolayer SRO using LEED experiment. LEED pattern of monolayer SRO measured (a) before, and (b) after post-annealing at 570 °C for 20 min. The LEED images were taken with an electron energy of 110 eV at 300 K.

We made the following changes to the revised manuscript.

(New figures) Figures A6 as Supplementary Fig. 11.

(Added, in the 9th paragraph) Possibilities for contributions from the surface states of STO and SRO layers to the spectroscopic results are discussed in **Supplementary Note 1**.

(Added, in Supplementary information) Supplementary Note 1: possible contributions from the surface states to the spectroscopic results

(**Added**, in Methods section) This process makes the surface quality of the sample suitable for ARPES measurements (**Supplementary Fig. 11**).

3. The films were annealed at 570 C for 20 mins in the ARPES chamber before measurements – oxygen vacancy creation is a possibility under these conditions. How did the authors rule out contributions from oxygen vacancies?

The reviewer raised a valid concern. This can be an issue if the annealing is not properly done. For such reason, the annealing condition has been carefully studied and we used the established annealing condition for all the samples before ARPES measurements. Therefore, the possible oxygen vacancy formations in SRO should not be an issue for the observed metal-insulator transition. Here are more details on the annealing condition.

For ARPES measurements of ultrathin films, we followed the annealing method reported in B. Sohn *et al.*, Nature Communications **12**, 6171 (2021). The films were annealed before ARPES measurements to obtain clean surfaces. In order to find the optimal annealing condition, temperature-dependent post-annealing was performed and photoemission spectra were taken. It was found that the optimized post-annealing condition was around 600 °C. Degradation of SRO after annealing over 600 °C was reported in same study. Therefore, we post-annealed SRO at 570 °C to avoid damage from accidental temperature fluctuation. The quasi-particle peak intensity to the high-binding peak intensity ratio (I_{QP}/I_{HB}) was the largest under the condition (**Figure A7**). It was found that SRO films annealed under the optimized condition had lower resistivities compared to as-grown SRO films. Note that the SRO films with oxygen vacancies are expected to have higher resistivities [E. K. Ko *et al.*, Advanced Functional Materials **30**, 20201486 (2020)]. Most importantly, K-dosing on annealed films showed very sharp quasi-particle peaks. This observation clearly indicates that properly annealed films have insignificant amount of oxygen vacancies (**Figure A8**) [Sohn *et al.*, Nature Communications **12**, 6171 (2021)]. Therefore, oxygen vacancy effect can be ruled out for SRO films annealed under the optimized condition.

Figure A7. Post-annealing temperature-dependent photoemission spectra from 20 u.c. SRO thin films on a STO (001) substrate near the Γ point. (a) Spectra before post-annealing (grey) and after post-annealing at 300 °C (black), 400 °C (red), 500 °C (blue), 600 °C (green), and 700 °C (orange). After annealing, a QP emerges. The intensity of QP increases with post-annealing temperatures but it eventually becomes weak after annealing at 700 °C. (b) The ratio of QP intensity in (a). r is defined as I_{QP} / I_{HB} , where I_{QP} is the intensity between E_F and $E_F - 0.1$ eV, and I_{HB} the integrated intensity between $E_F - 0.45$ eV and $E_F - 0.55$ eV. After post-annealing, r increases until 600 °C but decreases again when the SRO film is annealed at 700 °C, at which oxygen vacancies are believed to be created. The figure and caption are adopted from [B. Sohn *et al.*, Nature Communications **12**, 6171 (2021)].

Figure A8. K Dosing-dependent incoherent-to-coherent crossover in monolayer SRO (a) Fermi surface maps of pristine and K-dosed monolayer SROs measured at 10 K. (b) Band dispersions of pristine and K-dosed monolayer SROs along the $k_y = -0.2 \text{\AA}^{-1}$ line (red dotted line in a). (c) EDCs from pristine and K-dosed monolayer SRO films near the X point normalized by $E = -0.6 \text{ eV}$. With K dosing, a quasi-particle peak reappears near the Fermi level, while the hump peak in the high-binding energy region disappears, as marked by inverted triangles. 'K-dosed $\times 2$ ' indicates twice the dosing amount of 'K-dosed'. The figure and caption are adopted from [B. Sohn *et al.*, Nature Communications **12**, 6171 (2021)].

We have the following sentence in the revised manuscript to address the issue with a reference.

(**Added**, in Methods section) It was previously shown that oxygen vacancy effects are negligible for SRO films annealed under the optimized condition²⁵.

4. There is a lot of published work on orbitally controlled physics in oxides: For example, <https://www.science.org/doi/10.1126/science.1149338>; <https://journals.aps.org/prl/abstract/10.1103/PhysRevLett.94.156402>; <https://www.nature.com/articles/nphys2733>. I feel that the authors; <https://www.nature.com/articles/s41467-019-08472-y>. I suggest that this be captured – The introduction sounds like strain-controlled crystal field splitting and the associated orbital occupancy changes is a new approach. While this is a still a useful addition, a discussion about some of the previous approaches including charge transfer and strain could be useful.

We appreciate the reviewer's suggestion for citation of previous works on the orbital occupancy controls via strain engineering. The papers report results of experimental and theoretical studies on the strain-controlled crystal field splitting.

(**New references**) refs. 21-24 are newly cited in the introduction of the revised manuscript.

5. A few minor suggestions:

1) Line 59 – 61: t_{2g} orbital degeneracy is lifted for both compression and elongation – not just elongation.

The reviewer is right. We revised the sentence as follows:

(**Original** in the 3rd paragraph) For instance, the three t_{2g} orbitals split into d_{xy} and $d_{xz/yz}$ levels if the oxygen octahedron is elongated along the in-plane directions ($c < a$).

(**Revised**) For instance, the three t_{2g} orbitals split into d_{xy} and $d_{xz/yz}$ levels if the oxygen octahedron is elongated or compressed.”

2) Line 95: ‘...strain engineering to elongate oxygen octahedra..’ – along which direction?

We revised the sentence as follows:

(Original) in the 7th paragraph) We used strain engineering to elongate the oxygen octahedra in SRO monolayers.

(Revised) We used strain engineering to compress the oxygen octahedra along the out-of-plane direction in SRO monolayers.

3) Line 137-139, Isn't the statement: "the transition from d_{xz} to d_{xy} (or d_{yz}) not allowed" only true in the absence of any p - d hybridization?

The reviewer is right that the d - d transition is not allowed only when p - d hybridization is absent. This was meant to state that the transition is *small*. The strength of p - d hybridization can be estimated based on the orbital geometry. For example, there can be strong hybridization between $p_{x/y}$ of equatorial oxygen ion – d_{xy} of ruthenium ion while the hybridization between p_z – d_{xy} is small due to the orbital geometry. In the case of d_{xz} orbitals, they have a strong hybridization with p_z orbital of equatorial oxygen ion, and with p_x orbital of apical oxygen ion, but a small hybridization with $p_{x/y}$ orbital of equatorial oxygen ion. Therefore, the transition from d_{xy} to d_{xz} always involves a small hybridization and thus should be small due to the small orbital overlap.

We revised the sentence as follows:

(Original) in the 10th paragraph) Specifically, the transition from d_{xz} to d_{xy} (or d_{yz}) not allowed, given that there is only a small overlap between the corresponding orbitals.

(Revised) Specifically, the transition from d_{xz} to d_{xy} (or d_{yz}) will be small, given that there is only a small overlap between the corresponding orbitals.

4) Lines 177-179: "The MIT occurs at a strain of approximately +0.2%" I think that the data does not support this statement. The MIT could be anywhere between -0.5% and +0.2%.

We agree with the reviewer that the MIT could occur between -0.5% and +0.2%. We revised our claim accordingly.

(Original in the 14th paragraph) These strain-dependent DOSs at E_F in the SRO monolayer indicate that MIT occurs at a strain of approximately +0.2%, which agrees well with the optical spectra of the SRO monolayer **(Fig. 3g, h)**.

(Revised) These strain-dependent DOSs at E_F in the SRO monolayer indicate that MIT occurs at a strain between -0.5% and +0.2%, which agrees well with the optical spectra of the SRO monolayer **(Fig. 3g, h)**.

Reviewer #2

The authors of this manuscript report on a metal insulator transition driven by orbital selective occupancy of orbitals in strained monolayer SrRuO₃ films. The transition is controlled by the degree of tetragonal distortion of the unit cell imposed by growing the monolayers on top of a variety of substrates that span a range between compressive to tensile stress. Based on ellipsometric and ARPES experiments, the authors infer a Mott transition taking place in the d_{xz}/d_{yz} subset as the tensile strain grows sufficiently. The latter is ruled by strong correlations dictated by the ratio between Hund's coupling J and the U correlation energy.

To obtain reliable results, the authors devise a way to control the field splitting due to the tetragonal distortion, by developing a symmetry-preserving strain engineering technique (i.e., by inserting a STO layer between substrate and SrRuO₃ monolayer). The success of this strategy was confirmed by LEED diffraction patterns and transmission electron microscopy. This guaranteed that the relevant parameter was the c/a distortion and make reliable the comparison between the samples with different strain state.

They subsequently used ellipsometry over a wide range of frequencies and identified three main peaks. The authors propose specific transitions between electronic states in the t_{2g} manifold, ruled by parameters U and J . They assumed that electrons are transferred to neighboring sites in the lattice by interacting with light, which I think is a reasonable and justified hypothesis. Based on the evolution of the spectral weight with strain, they interpreted the results as coming from the relative shift in energy of d_{xy} and d_{xz}/d_{yz} orbitals induced by tetragonal distortions and electronic correlations.

The authors used data from ARPES to determine the orbital character of states below the Fermi energy. They checked that orbitals closer to E_F are d_{xy} , while those down in energy are of d_{xz}/d_{yz} character.

I think that this is an interesting work that combines efforts of materials science with fundamental insights into the physics of correlated systems. I have a couple of comments:

We thank the reviewer for the comments. We have answered both of the questions as follows.

1. First, I suggest making a bit clearer the discussion about the orbital hierarchy inferred from ARPES. The experiments show that under positive strain, where $c/a < 1$, the $d_{xz/yz}$ orbitals are further away from E_F than d_{xy} orbitals. This is not compatible with an orbital hierarchy fully determined from c/a tetragonality, since for $c/a < 1$ one should expect d_{xy} states to be lower in energy (as depicted in Fig. 1b). This observation could be used by the authors to reinforce their viewpoint. A Mott transition with UHB/LHB splitting in the d_{xy}/d_{xz} sector would be compatible with ARPES observations, which otherwise could not explain by pure distortion effects. On the other, I suggest the authors to discuss the c/a parameter as extracted from STEM and include this discussion in the manuscript.

We thank the reviewer for helping us make the discussion clearer. As the reviewer mentioned, the $c/a < 1$ makes the d_{xy} orbital level lower than the $d_{xz/yz}$ level when only the tetragonal distortion effect is considered. Therefore, d_{xy} is fully filled while $d_{xz/yz}$ is half-filled when $U = J = 0$ (as illustrated in the middle figure of **Figure B1**). In real SRO (with sizable U and J), the half-filled $d_{xz/yz}$ will open a Mott gap (as shown in the right figure in **Figure B1**). This makes $d_{xz/yz}$ orbitals farther away from E_F than the d_{xy} orbital. In other words, in addition to the pure structural distortion, effects of U and J should be considered to elaborate on the observed electronic structures in SRO as the reviewer correctly pointed out.

Figure B1. Schematic of the band configuration of SRO. With a small Δ_t , partially-filled $d_{xy/xz/yz}$ orbitals form metallic bands. With a large Δ_t , fully-filled d_{xy} becomes band insulating while half-filled $d_{xz/yz}$ becomes Mott insulating with a sizable on-site Coulomb interaction (U) and Hund's rule coupling (J). (**Figure 1b**)

To make it clearer, we made the following changes.

(New Labels) “without U & J ” and “with U & J ” for the middle and right figures of **Figure 1b**, respectively (**Figure B1**).

(Added, 4th paragraph) This crystal field splitting effect (i.e. with negligible U/D) makes d_{xy} level lower than that of $d_{xz/yz}$.

(Original in the 4th paragraph) When U/W is large (W is the bandwidth), the $d_{xz/yz}$ bands can further experience a Mott transition¹⁴

(Revised) When U/D is large (D is the half-bandwidth), the $d_{xz/yz}$ bands can further experience a Mott transition¹⁴, as schematically shown in the last band configuration of **Fig. 1b**.

(Added, 4th paragraph) The relative position of $d_{xz/yz}$ can be a good indication of how U and the crystal field play roles in determining electronic structures in of SRO.

Figure B2 shows the extracted lattice constants from HAADF images for films on LSAT(001) and SAGT(001) substrates. We used the A-site cationic sublattice to obtain lattice constants. The

LSAT(001) and SAGT(001) substrates induce -1.4% and +0.2% of strain compared to the bulk SRO lattice constant, respectively. Left graphs of **Figures B2c** and **B2d** plot in-plane lattice constants as a function of the atomic layer number, where a_{sub} is the in-plane lattice constant of substrate. We can conclude that our films are fully-strained to the substrates. Right graphs of **Figures B2c** and **B2d** depict c/a_{sub} . The c/a_{sub} value of SRO monolayer on LSAT(001) is 102.8% whereas that of SRO monolayer on SAGT(001) is 101.8%, confirming that tensile strain leads to a smaller c/a value.

Although the trend that tensile strain reduces the c/a_{sub} value is consistent with the expectation, the value from the STEM analysis is not exactly same as the expected value. A potential source for the difference is that the STO capping layer (for the protection of the SRO layer during STEM measurements) could affect the structure of the underneath SRO layer. In addition, assigning precise position at the interfaces can be challenging due to cation mixing or step terraces, leading to large error bars. Therefore, only the qualitative aspect of the result should be taken.

Figure B2 Lattice constant analysis from STEM results for STO(10 u.c.)/SRO(1 u.c.)/STO(10 u.c.) on LSAT(001) and SAGT(001). a,b, STEM-HAADF images of STO(10 u.c.)/SRO(1 u.c.)/STO(10 u.c.) on LSAT(001) (a) and SAGT(001) (b). c,d, Lattice constants as a function of the atomic row number. The in-plane lattice constants show both SRO and STO layers are fully strained to the substrates. c/a_{sub} shows that tensile strained SRO and STO layers have smaller c/a_{sub} values compared to the compressive strained films.

2. My second comment is about the role of spin-orbit coupling (SOC), which is relevant in 4d ions (SOC is in the range of 0.2-0.3 eV for these ions) but is not discussed at all in the manuscript. The physics of SOC in t_{2g} systems is described in some references (see, e.g., Streltsov et al., Phys. Rev X 10, 031043 (2020) or Khomskii and Streltsov, Chem. Rev. 121, 5, 2992 (2021)). The Δ_t “crystal field” parameter, originated by the c/a distortion, is in the range of

a few hundreds of meV (as inferred from EDCs data in Fig. S5). This is not far from energy scales of SOC in 4d or from exchange coupling energy. In particular, my question is whether the LHB band (inferred from ARPES experiments, see Figures 5 or S5) is split because of SOC. Maybe the energy resolution is not enough to discriminate the splitting, but in any case, I suggest the authors to discuss the role of spin-orbit coupling in the physics of the orbital-selective phase transitions discussed in this manuscript.

We thank the reviewer for the suggestion. SOC can indeed play important roles in 4d t_{2g} orbital physics as the reviewer pointed out. The energy scale of SOC in ruthenates is around 0.1 ~ 0.2 eV [Z. Fang *et al.*, Physical Review B **69** 045116 (2004), G. Zhang *et al.*, Physical Review B **95** 075145 (2017)], which is not negligible compared to the energy scale of the crystal field and the exchange interaction J . SOC in ruthenates can change fermiology, especially close to the degeneracy points [M. Kim *et al.*, Physical Review Letters **120** 126401 (2018)]. It can also promote band degeneracy lifting for nodal structures, resulting in large Berry curvature [B. Sohn *et al.*, Nature Materials **20** 7512 (2021)]. A point to note is that, if SOC dominates the physics, orbital differentiation should not be observed since SOC promotes orbital mixing.

According to recent theoretical studies, the orbital polarization in ruthenates is mainly determined by the structural distortion, rather than SOC [G. Zhang *et al.*, Physical Review B **95** 075145 (2017); M. Kim *et al.*, Physical Review Letters **120** 126401 (2018)]. Since the observed orbital-selective phase transition in our work is mainly caused by the orbital polarization, we may deduce that the orbital-selective Mott transition is not significantly affected by the SOC.

As for the band splitting in the LHB the reviewer mentioned, it may not be experimentally observed because of the breadth of the incoherent LHB at the high binding energy.

We have the following sentence in the revised manuscript to address the issue.

(Added in the 12th paragraph) Note that existence of such orbital polarization alludes to an insignificant role of the spin-orbit coupling in the orbital dependent Mott transition in the ruthenate films³⁹.

Reviewer #3

The authors design and realize a smart way to tune the band structure of SrRuO₃ by growing thin films on top of different substrates avoiding distortions in the oxygen octahedra. This allows them to detect a metal insulator transition as a function of the strain induced by the substrate. The authors ascribe the transition to an orbital selective Mott transition due to Hund's physics. I have some concerns with several of their arguments and discussion:

We thank the reviewer for acknowledging the novelty of our work and also for the helpful comments. We responded to all the questions raised and revised the manuscript accordingly.

1. The phase diagram in Fig. 1a is confusing, and it seems to me partially wrong. As far as I understand the $J/4$ and $J = 0$ have to be compared with the negative crystal field splitting in Ref. 14 (figs. 1a and 3a). There are however several differences: (i) Why is the crystal field here given in eV? Ref.14 seems to be given in units of the half-bandwidth D . (ii) The $J=0$ boundaries do not show the same tendencies as in Fig.3a of Ref.14. (iii) Moreover plotting the $J=U/4$ and $J=0$ is confusing especially because they color the phase diagram following the $J/4$ lines. I suggest the authors plot the two values of Hund's coupling separately and be careful when plotting the lines.

We thank the reviewer for the very helpful comment. The phase diagram in **Figure 1** in the original manuscript can be confusing and mislead the readers. Therefore, we revised **Figure 1c** as the reviewer suggested (see **Figure C1**).

(i)

(Original in the Figure 1) The x -axis was given in crystal field splitting (Δ_t).

(Revised) The x -axis is now given in (crystal field splitting)/(half-bandwidth) (Δ_t/D).

(ii)

(Original in the Figure 1) The quantitative relationship between $J = 0$ boundary line and $J = U/4$ boundary line did not match the previous calculation result [L. Huang *et al.*, Physical Review B **86** 035150 (2012)].

(Revised) $J = 0$ boundary line is revised to show the same tendencies as in Figure 3a of Ref.14 [L. Huang *et al.*, Physical Review B **86** 035150 (2012)]. In this reference, when $\Delta_v/D = 0$, the U_c/D for $J = U/4$ line and $J = 0$ line are around 10 and 5.5, respectively. When $\Delta_v/D = 1$, the U_c/D for $J = U/4$ line and $J = 0$ line are around 2.2 and 4.2, respectively. $D = 1.0$ was used for the calculation.

(iii)

(Original in the Figure 1) Boundary lines for $J = U/4$ and $J = 0$ were plotted in single phase diagram.

(Revised) Phase diagrams with $J = U/4$ and $J = 0$ are plotted separately.

Figure C1 Phase diagram of U/D versus Δ_v/D . D is the half-bandwidth. Left figure is a phase diagram with $J = U/4$, while right figure shows a phase diagram with $J = 0$. **(Figure 1c)**

2. I think that the experimental MIT is robust. As far as I understand the authors claim that this MIT corresponds to the transition between a metal with both d_{xy} and d_{zx}/d_{yz} orbitals partially filled and an insulator with d_{xy} fully occupied and d_{zx}/d_{yz} half-filled. They argue that this is a proof of Hund's physics. While Hund's coupling increases the range of U/W in which this transition can happen, as observed in Fig.3a of Ref.14 this transition can also happen at $J = 0$. Therefore in my opinion the authors cannot claim that they observe Hund's physics.

We agree that the observation of the orbital-selective phase transition itself is not a direct evidence for the Hund's physics. A direct observation of Hund's physics is challenging because tuning J value is experimentally difficult. In this study, we suggest an indirect way to control the orbital occupancy for the study of phase transition in Hund's system. We wanted to address that the phase transition in 2D SRO can provide insight on the Hund's physics. We revised the sentences that can mislead the readers (please see below).

We made the following revisions:

(Original in the 2nd paragraph) The J value is usually determined by atomic physics, so control of its value is difficult without chemical substitution. To control orbital occupancy, earlier studies used doping and/or substitution with different chemical elements.

(Revised) Direct control of the J value is experimentally difficult as it is usually determined by atomic physics. On the other hand, an alternative but indirect approach to control orbital occupancy is possible via doping and/or substitution with different chemical elements.

(Original in the 4th paragraph) In this study, we investigated how Hund-driven phase transitions can occur in SrRuO₃ (SRO) films by artificially controlling the crystal field splitting.

(Revised) In this study, we investigated tuning of orbital occupancy in SrRuO₃ (SRO) ultrathin films by artificially controlling the crystal field splitting. It should be noted that Sr₂RuO₄ is well-known as Hund's metal, so 2D limit of SRO can provide insight on the Hund's physics⁷

(Original in the 2nd paragraph) Therefore, precise control of orbital occupancy without random chemical distribution is the key for experimentally investigating Hund-driven phase transitions.

(Revised) Therefore, precise control of orbital occupancy without random chemical distribution is the key for experimental investigation of phase transitions in Hund's systems.

What we would like to address was that the observed strain induced MIT shows a character of Hund's systems; as the reviewer mentioned, the range of U/D in which the MIT can happen is increased in Hund's systems. This is relevant to the ambivalent role of J in the system, where the critical U for opening a Mott gap (U_c) can be increased/decreased depending on the orbital occupancy. For example, U_c/D can vary between 2.2 and 10 for $J = U/4$, while it changes only

between 4.2 and 5.5 with $J = 0$ [L. Huang *et al.*, Physical Review B **86** 035150 (2012)]. In this sense, one of the characteristics of Hund's systems is that the crystal field-induced Mott gap opening is robust. If we rule out the effect of Hundness ($J = 0$), most physics will be determined solely by the U/D value.

In order to investigate the band-width change effect, we performed additional temperature-dependent ARPES experiments. According to previous reports [R. He *et al.*, Physical Review B **105** 064104 (2022); T. Riste *et al.*, Solid State Communications **88** 901 (1993); E. Heifets *et al.*, Journal of Physics: Condensed Matter **18** 4845 (2006)], the out-of-plane oxygen octahedral rotation (OOR) angle in STO decreases with increasing temperature. It will also decrease the OOR angle in SRO monolayers due to the hetero-interfacial coupling effect [J. M. Rondinelli *et al.*, MRS Bulletin **37** 261 (2012)], leading to an increase in the bandwidth. To see whether the bandwidth change affects the electronic structure of SRO, we measured temperature-dependent ARPES of SRO with +0.2 % of strain which is insulating at low temperature. As shown in **Figure C2**, the data exhibit a robust insulating phase between 6 and 160 K with the lower Hubbard band (LHB) of $d_{xz/yz}$ character. In other words, the Mott transition in this system cannot be induced by band-width change in tensile-strained film. Therefore, we believe the robust strain-induced Mott gap opening is an *indication* of a Hund's system (again, not an evidence as the reviewer pointer out).

Figure C2 Temperature dependent energy distribution curves (EDCs) and E - k dispersion of the SRO monolayer (+ 0.2 %) along the Γ -M line.

3. I think that the numbers of the interactions do not match well the arguments from the authors. If I understand the arguments by the authors, from Fig.3 they obtain $U - J \sim 1.2$ eV, $U + J \sim 2.4$ eV, thus $J \sim 0.6$ eV and $U \sim 1.8$ eV. But in Fig.5 the lower Hubbard band is centered around -1.6 eV. This value can be expected to be around $U/2$. Then I would expect U to be around 3.6 eV, much larger than the value obtained in Fig.3. Am I wrong with the interpretation of the results? Could the authors comment on this?

We thank the reviewer for bringing up the comparison between energy scales of electronic correlations and ARPES band peak positions.

First, the gap between lower Hubbard band (LHB) and upper Hubbard band (UHB) in our tensile-strained SRO monolayer can be differed from U when we include the effect of J . In the 2-orbital/2-electron half-filled case, the Mott gap increases to $U + J$ [A. Georges et al., *Annu. Rev. Condens. Matter Phys.* **4** 137 (2013)].

Second, as the reviewer pointed out, the energy position of the LHB measured by ARPES is about 1.6 eV below E_F . In the case of a single-band system, this band will split symmetrically by $(U+J)/2$ around E_F . However, in multi-orbital systems, the presence of other band makes the energy levels of LHB and UHB asymmetric around E_F . In the case of monolayer SRO, E_F will be located between the d_{xy} band (not LHB) and UHB, which are the top of valence band and the bottom of conduction band, respectively. In an analogous system of Ca_2RuO_4 which has the same orbital hierarchy and Mott insulating behavior, DMFT results show that UHB is closer to E_F than LHB [S. Ricco *et al.*, *Nature Communications* **9** 4535 (2018); D. Sutter *et al.*, *Nature Communications* **8** 15176 (2017)] (**Figure C3**).

Figure C3 DMFT results of Ca_2RuO_4 . **a**, calculated band structure which has d_{xy} orbital character (left) and $d_{xz/yz}$ orbital character (right) based on DMFT. The figure is adopted from [D. Sutter *et al.*, Nature Communications **8** 15176 (2017)]. **b**, ARPES and DMFT result of Ca_2RuO_4 . The figure adopted from [S. Ricco *et al.*, Nature Communications **9** 4535 (2018)].

Lastly, the extraction of values from optical results should be taken with caution. The transition peak positions of optical spectra could be lowered due to the excitonic binding energy. The formation of excitons during optical processes could significantly lower the transition peak position [J. S. Lee *et al.*, Physical Review B **64** 245107 (2001)]. Therefore, the extracted U value from optical peak positions can be underestimated.

We have the following sentences in the revised manuscript to address the issue.

(Added in the 11th paragraph) The extracted U values from the optical results could be underestimated due to formation of excitons during optical processes³⁷.

(Added in the 17th paragraph) More details on the energy positions of LHB and UHB are discussed in **Supplementary Note 3**.

(Added, in Supplementary information) Supplementary Note 3: the energy positions of LHB and UHB

4. Can the authors give more detailed information about the effective tight-binding that they obtain from the Wannierization for the t_{2g} orbitals? In particular how much does it change the crystal field with the strain? Is there any change in the bandwidth of the non interacting band? And particularly for the d_{zx} and d_{yz} subspace? Also, which value of U do the authors use in the calculations? How large is the orbital dependent bandwidth? I think that I have not seen any of this information either in the main text or in the supplemental material.

We thank the reviewer for raising the point. We left out detailed but important information on the theory part in the original manuscript. **Figure C4** shows detailed band structures obtained from the Wannierization for the t_{2g} orbitals. In addition, strain dependent crystal field splitting ($\Delta_t = E_{xz/yz} - E_{xy}$) is shown in **Figure C5**. Electron hopping terms for d_{xy} - d_{xy} and d_{xz} - d_{xz} (or d_{yz} - d_{yz}) are shown in **Figure C6**. In this study, we used $U = 2.7$ eV and $J = 0.45$ eV ($J = U/6$).

Figure C4 Electronic structure obtained from Wannierization for the t_{2g} orbitals for various strain values.

Figure C5 Crystal field splitting depending on strain. Tensile strain gives larger crystal field splitting.

Figure C6 Orbital dependent hopping energy.

We made the following changes to the manuscript

(Original, in the 12th paragraph) Dynamic mean-field theory (DMFT) calculations with $J = U/6$ also revealed that the filling of d_{xy} (d_{xz}/d_{yz}) orbitals increased (decreased) with an increase in strain (**Fig. 3h**).

(Revised) Dynamic mean-field theory (DMFT) calculations with $J = U/6$ ($U = 2.7$ eV, $J = 0.45$ eV) also revealed that the filling of d_{xy} (d_{xz}/d_{yz}) orbitals increased (decreased) with an increase in strain (**Fig. 3h**).

(New figures) Figure C4 and **Figure C6** are included in the **Supplementary** as **Fig. 13** and **Fig. 14**.

5. Can the authors give in Fig. 3h the DMFT electron filling for the cases with strain 1.7 and 2.5%. This is relevant for the interpretation of the ARPES results.

We performed DMFT electron filling calculation with + 1.7 % and +2.5 % of strain. The results in **Figure C7** shows that further tensile strain makes d_{xy} fully filled, which is consistent with the ARPES results.

Figure C7 Strain-dependent electron filling calculated via DMFT with $U = 2.7$ eV and $J = 0.45$ eV.

We made the following changes to the manuscript.

(New figure) Figure C7 is included in the **Supplementary** as **Fig. 6**.

(Original, in the 12th paragraph) When the strain reaches +0.2%, d_{xy} becomes fully-filled, and $d_{xz/yz}$ becomes half-filled

(Revised) When the strain is +0.2% or higher, d_{xy} is fully-filled while $d_{xz/yz}$ is half-filled

(Supplementary Fig. 6).

6. Due to all these concerns, in my opinion the manuscript, at least in its present version is not suitable for publication in Nature Comm.

We sincerely appreciate the reviewer's comments which greatly helped us improve the manuscript. Based on the reviewer's comment, we revised our manuscript. **Figure 1** and introduction were revised. We performed additional experiments and calculations as well. We believe that our revised manuscript is more solid and accurate.

Part B: List of changes

All the changes in the revised Main text, Figures, References, and Supplementary information are listed as follows:

1. In the main text, 2nd paragraph,
“The J value is usually determined by atomic physics, so control of its value is difficult without chemical substitution. To control orbital occupancy, earlier studies used doping and/or substitution with different chemical elements^{19,20}.” was changed to “**Direct control of the J value is experimentally difficult as it is usually determined by atomic physics. On the other hand, an alternative but indirect approach to control orbital occupancy is possible via doping and/or substitution with different chemical elements^{19,20}.**”
2. In the main text, 2nd paragraph,
“Therefore, precise control of orbital occupancy without random chemical distribution is the key for experimentally investigating Hund-driven phase transitions.” was changed to “**Therefore, precise control of orbital occupancy without random chemical distribution is the key for experimental investigation of phase transitions in Hund’s systems.**”
3. In the main text, 3rd paragraph,
“We propose that crystal field splitting can be controlled with a suitable experimental approach for observing Hund-driven phase transitions with negligible impurity problems.” was changed to “**We propose that crystal field splitting can be controlled with a suitable experimental approach to observation of phase transitions in Hund’s systems with negligible impurity problems.**”
4. In the main text, 3rd paragraph,
“For instance, the three t_{2g} orbitals split into d_{xy} and $d_{xz/yz}$ levels if the oxygen octahedron is elongated along the in-plane directions ($c < a$).” was changed to “**For instance, the**

three t_{2g} orbitals split into d_{xy} and $d_{xz/yz}$ levels if the oxygen octahedron is elongated or compressed.”

5. In the main text, 4th paragraph,

“In this study, we investigated how Hund-driven phase transitions can occur in SrRuO₃ (SRO) films by artificially controlling the crystal field splitting.” was changed to “**In this study, we investigated tuning of orbital occupancy in SrRuO₃ (SRO) ultrathin films by artificially controlling the crystal field splitting. It should be noted that Sr₂RuO₄ is well-known as Hund’s metal, so 2D limit of SRO can provide insight on the Hund’s physics⁷**”

6. In the main text, 4th paragraph,

“When U/W is large (W is the bandwidth), the $d_{xz/yz}$ bands can further experience a Mott transition¹⁴, which is displayed schematically with the last band configuration of **Fig. 1b**.” was changed to “**This crystal field splitting effect (i.e. with negligible U/D) makes d_{xy} level lower than that of $d_{xz/yz}$. When U/D is large (D is the half-bandwidth), the $d_{xz/yz}$ bands can further experience a Mott transition¹⁴, as schematically shown in the last band configuration of **Fig. 1b**. The relative position of $d_{xz/yz}$ can be a good indication of how U and the crystal field play roles in determining electronic structures in of SRO.**”

7. In the main text, 7th paragraph,

“We used strain engineering to elongate the oxygen octahedra in SRO monolayers.” was changed to “**We used strain engineering to compress the oxygen octahedra along the out-of-plane direction in SRO monolayers.**”

8. In the main text, 8th paragraph,

“On the other hand, no sample showed (0.5 m, 0.5 n) peaks, indicating that OOR did not occur along the in-plane axis.” was changed to “**On the other hand, no sample showed (m+0.5, n) and (m, n+0.5) peaks, indicating that OOR did not occur along the in-plane axis.**”

9. In the main text, 9th paragraph,

“The analysis of lattice constants from the STEM results shows coherent strain state (**Supplementary Fig. 4**).” was added.

10. In the main text, 9th paragraph,

“Possibilities for contributions from the surface states of STO and SRO layers to the spectroscopic results are discussed in **Supplementary Note 1**.” was added.

11. In the main text, 10th paragraph,

“Specifically, the transition from d_{xz} to d_{xy} (or d_{yz}) not allowed, given that there is only a small overlap between the corresponding orbitals.” was changed to “**Specifically, the transition from d_{xz} to d_{xy} (or d_{yz}) will be small, given that there is only a small overlap between the corresponding orbitals.**”

12. In the main text, 11th paragraph,

“The extracted U values from the optical results could be underestimated due to formation of excitons during optical processes.” was added.

13. In the main text, 12th paragraph,

“($U = 2.7$ eV, $J = 0.45$ eV)” was added.

14. In the main text, 12th paragraph,

“When the strain reaches +0.2%, d_{xy} becomes fully-filled, and $d_{xz/yz}$ becomes half-filled” was changed to “**When the strain is +0.2% or higher, d_{xy} is fully-filled while $d_{xz/yz}$ is half-filled (**Supplementary Fig. 6**).**” was added.

15. In the main text, 12th paragraph,

“Note that existence of such orbital polarization alludes to an insignificant role of the spin-orbit coupling in the orbital dependent Mott transition in the ruthenate films³⁹ .” was added.

16. In the main text, 14th paragraph,

“These strain-dependent DOSs at E_F in the SRO monolayer indicate that MIT occurs at a strain of approximately +0.2%, which agrees well with the optical spectra of the SRO monolayer (Fig. 3g, h).” was changed to “These strain-dependent DOSs at E_F in the SRO monolayer indicate that MIT occurs at a strain between -0.5% and +0.2%, which agrees well with the optical spectra of the SRO monolayer (Figs. 3g and 3h).”

17. In the main text, 17th paragraph,

“More details on the energy positions of LHB and UHB are discussed in **Supplementary Note 3.**”

18. In the main text, 18th paragraph,

“Since the energy center position of the Hubbard bands varied little with strain, ζ should be close to Δ_t .” was changed to “Since the LHB energy varies little with the strain, the change in the ζ value should be from the change in Δ_t .”

19. In the Methods section, 3rd paragraph,

“The heterostructures are fully strained to substrates as shown in **Supplementary Fig. 10.**” was added.

20. In the Methods section, 3rd paragraph,

“This process makes the surface quality of the sample suitable for ARPES measurements (**Supplementary Fig. 11**). It was previously shown that oxygen vacancy effects are negligible for SRO films annealed under the optimized condition²⁵” was added.

21. In the Methods section, 6th paragraph,

“**Supplementary Fig. 13 and Fig. 14** show the detailed band structure obtained from the Wannierization for the t_{2g} orbitals and the electron hopping term of d_{xy} - d_{xy} and d_{xz} - d_{xz} (or d_{yz} - d_{yz}).” was added.

22. **Figure 1** is revised as shown below:

23. References [21-24, 39] were added.

21. Chakhalian, J., Freeland, J. W., Habermeier, H. U., Cristiani, G., Khaliullin, G., Van Veenendaal, M. & Keimer, B. Orbital reconstruction and covalent bonding at an oxide interface. *Science*. **318**, 5853 (2007).

22. Khomskii, D. I. & Mizokawa, T. Orbitaly induced Peierls state in spinels. *Phys. Rev. Lett.* **94**, 156492 (2005).

23. Aeukuri, N. B., Gray, A. X., Drouard, M., Cossale, M., Gao, L., Reid, A. H., Kukreja, R., Ohldag, H., Jenkins, C. A., Arenholz, E., Roche, K. P., Durr, H. A., Samant, M. G & Parkin, S. S. P. *Nature Physics*. **6**, 661 (2013).

24. Liao, Z., Skoropata, E., Freeland, J. W., Guo, E. J., Desautels, R., Gao, X., Sohn, C., Rastogi, A., Ward, T. Z., Zou, T., Charlton, T., Fizsimmons, M. R. & Lee, H. N. *Nat. Commun.* **10**, 589 (2010).

39. Kim, M., Mravlje, J., Ferrero, M., Parcollet, O. & Georges, A. Spin-orbit coupling and electronic correlations in Sr_2RuO_4 . *Phys. Rev. Lett.* **120**, 126401 (2018)

24. In supplementary information, **Supplementary Fig. 4, 6, 10, 11, 13, 14** were added.

25. In supplementary information, **Supplementary Note 1, 3** were added.

REVIEWERS' COMMENTS

Reviewer #1 (Remarks to the Author):

While I am happy with the efforts that the authors have put in with respect to the new experiments and data, I am not entirely convinced with the arguments regarding the annealing of the samples at high temperatures. Instead of showing data from a previously published work, it would have been more useful if the authors showed photoemission data at lower annealing temperatures for these same samples.

Regarding a question on previously published work on orbitally controlled phase transitions in oxides, it is important that authors contextualize the current work in the light of this previously published literature and why Hund's physics is invoked in this work. Simply citing the references might not be sufficient.

Reviewer #2 (Remarks to the Author):

The authors answered satisfactorily my questions regarding the role of spin-orbit coupling on the physics of the t_{2g} orbital hierarchy in the SRO ultrathin films. They also made a clear statement about my comments on the orbital hierarchy inferred from ARPES. In the revised version, the authors have addressed in full detail and exhaustively all comments from Referees. As I mentioned in my previous correspondence, the work brings fundamental insights into the physics of correlated systems, and I think it deserves publication in the journal.

Reviewer #3 (Remarks to the Author):

The authors have carefully addressed the comments and suggestions from the referees, made the changes in the manuscript and included extra information to clarify the issues raised. The present version is clearly improved.

There is one issue, that I raised previously, which, even if it has been improved, it is still confusing and should be changed before publication. It refers to figure 1c and the text which refers to this figure (lines 79-84 in the version that I have received). Regarding the figure: The authors point to the

blue area in the $J=0$ panel as Hund metal state. There is clearly no Hund metal at $J=0$! So this should be changed.

Even in the $J=U/4$ figure where a Hund metal does exist, the authors should be careful. Not all the blue area is a Hund metal, but only a part. For example, at very small U the system will be very weakly correlated and not a Hund metal. To give the authors an idea, I suggest them to see Fig.2 in PRB 92, 075136 (2015). Their case is close to panel (d) in this figure at $J=U/4$. Only the violet region in such figure is a Hund metal. In the blue area of the authors' figure they have some part that would be "violet" (i.e. strongly correlated and with well formed moments but metallic, i.e. Hund metal) but some other part would be "yellowish" (moderate correlations). Without information on the quasiparticle weight it is difficult to say exactly where it is the crossover in the authors figure, but they will for sure not have a Hund metal for very small U . So they have to discuss this more carefully.

My suggestion for the paragraph, lines 79-84, it would be to say that for $J=0$ the transition induced by crystal field is not very easy to detect because it happens in a small region of U and Hund's coupling facilitates this transition in a much wider interaction range. I find better not to mention the Hund's metal when they discuss the transition from a metallic state to another metallic state, but when they go from the metallic state to the insulating state.

Once this issue is clarified the manuscript will be suitable for publication.

We would like to thank the reviewers for their useful comments and helpful suggestions, which improved our manuscript. Please find point-by-point responses to the comments and a list of changes made in the revised manuscript.

Responses to the comments

Reviewer #1

While I am happy with the efforts that the authors have put in with respect to the new experiments and data, I am not entirely convinced with the arguments regarding the annealing of the samples at high temperatures. Instead of showing data from a previously published work, it would have been more useful if the authors showed photoemission data at lower annealing temperatures for these same samples.

We appreciate the reviewer's constructive comment based on his/her concern on the quality of our SrRuO₃ monolayer films during the annealing process at 570 °C. To address the sample quality during the annealing process, some of us previously reported that 20 u.c. of SrRuO₃ films have high quality suitable for photoemission after being annealed at 600 °C [B. Sohn et al., Nature Communications 12, 6171 (2021)], which is close to 570 °C used in this study. However, we are mainly dealing with monolayers of SrRuO₃ and thus it is worthwhile reconfirming the effect of the annealing process for monolayer systems as the reviewer suggested.

We performed annealing-temperature dependent photoemission measurements of SrRuO₃ monolayer (i.e., SrRuO₃ (1 u.c.)/SrTiO₃ (10 u.c.)/SrRuO₃ (4 u.c.)/SrTiO₃ (10 u.c.) on LSAT(001) (- 1.4 % strain)) (**Figure A1**). The results from SrRuO₃ monolayer samples were consistent with the previously reported results from 20 u.c. films. Namely, after annealing at between 400 and 600 °C, monolayer SrRuO₃ films show high photoemission intensity near the Fermi level, demonstrating the high quality of our films. These additional experiments at various annealing temperatures demonstrates that our original argument based on the 570 °C annealed SrRuO₃ monolayer is still valid. **Figure A1** is included in the **Supplementary Information** as **Fig. 15**.

Figure A1. Annealing-temperature dependent photoemission spectra of SrRuO₃ (1 u.c.)/SrTiO₃ (10 u.c.)/SrRuO₃ (4 u.c.)/SrTiO₃ (10 u.c.) on LSAT(001). The samples were annealed at 300, 350, 400, 500, 600, and 700 °C in an ultrahigh vacuum ($< 1.0 \times 10^{-10}$ Torr) before photoemission measurements. The quasiparticle peak intensity ratio defined as $r = I(E - E_F = -0.05 \text{ eV})/I(E - E_F = -0.4 \text{ eV})$ is shown on the right.

Regarding a question on previously published work on orbitally controlled phase transitions in oxides, it is important that authors contextualize the current work in the light of this previously published literature and why Hund's physics is invoked in this work. Simply citing the references might not be sufficient.

We appreciate the reviewer's constructive suggestion. The main difference between previous works and our work is the material systems. Previous studies on orbitally controlled phase transitions [ref. 22, 23, 24 in the main text] were mostly focused on 3d transition metal oxides where physics is dominantly driven by the large U value. In this work, we orbitally controlled a two-dimensional 4d transition metal oxide (i.e. 2D ruthenate) which is a well-known Hund's system. Following the reviewer's suggestion, we added a few sentences highlighting our system in the introduction and discussion of the revised manuscript.

(3rd paragraph, 9-14 line in the main text) “There have been previous experimental studies that demonstrated the manipulation of orbital polarization²²⁻²⁴. They include the charge transfer at cuprate-manganite interfaces²², metal-insulator transition in VO₂²³, and nickelate-cuprate heterostructures²⁴. However, the impact of J in such orbital polarization changes has not been considered. Here, our objective is to explore the interplay between Δ_t and J by controlling the orbital polarization in a widely accepted Hund’s system.”

(20th paragraph, 6-8 line in the main text) “Note that J and crystal field energy scales compete each other in terms of the orbital polarization. Therefore, the observed orbital-selective phase transition induced by the crystal field tuning in Hund’s system will provide an insight on the role of J .”

Reviewer #2

The authors answered satisfactorily my questions regarding the role of spin-orbit coupling on the physics of the t_{2g} orbital hierarchy in the SRO ultrathin films. They also made a clear statement about my comments on the orbital hierarchy inferred from ARPES. In the revised version, the authors have addressed in full detail and exhaustively all comments from Referees. As I mentioned in my previous correspondence, the work brings fundamental insights into the physics of correlated systems, and I think it deserves publication in the journal.

We thank the reviewer for his/her comment.

Reviewer #3

The authors have carefully addressed the comments and suggestions from the referees, made the changes in the manuscript and included extra information to clarify the issues raised. The present version is clearly improved.

There is one issue, that I raised previously, which, even if it has been improved, it is still confusing and should be changed before publication. It refers to figure 1c and the text which refers to this figure (lines 79-84 in the version that I have received). Regarding the figure: The authors point to the blue area in the $J=0$ panel as Hund metal state. There is clearly no Hund metal at $J=0$! So this should be changed.

Even in the $J=U/4$ figure where a Hund metal does exist, the authors should be careful. Not all the blue area is a Hund metal, but only a part. For example, at very small U the system will be very weakly correlated and not a Hund metal. To give the authors an idea, I suggest them to see Fig.2 in PRB 92, 075136 (2015). Their case is close to panel (d) in this figure at $J=U/4$. Only the violet region in such figure is a Hund metal. In the blue area of the authors' figure they have some part that would be "violet" (i.e. strongly correlated and with well formed moments but metallic, i.e. Hund metal) but some other part would be "yellowish" (moderate correlations). Without information on the quasiparticle weight it is difficult to say exactly where it is the

crossover in the authors figure, but they will for sure not have a Hund metal for very small U . So they have to discuss this more carefully.

We sincerely thank the reviewer for the corrections and apologize for the confusion. As she/he suggested, we changed **Figure 1c** as shown below.

1. $J = 0$, there is no Hund's metallic phase. Blue region is indicated as metal.
2. $J = U/4$, the Hund's metallic phase is shown by the purple region, which is located between the metallic and insulating phases.

My suggestion for the paragraph, lines 79-84, it would be to say that for $J=0$ the transition induced by crystal field is not very easy to detect because it happens in a small region of U and Hund's coupling facilitates this transition in a much wider interaction range. I find better not to mention the Hund's metal when they discuss the transition from a metallic state to another metallic state, but when they go from the metallic state to the insulating state. Once this issue is clarified the manuscript will be suitable for publication.

We greatly appreciate reviewer's suggestions.

We revised the sentences in the introduction in the revised manuscript as

(5th paragraph, 3-9 line in main text) “With $J = 0$, it can be challenging to detect the Mott transition induced by the crystal field because it only happens in a small range of U . On the other hand, J facilitates such transition in a much wider interaction range. Therefore, the system with a sizable J experiences orbital-selective phase transitions as Δ_t varies.”

“For a small U/D , increasing Δ_t induces a transition from a metallic state into another metallic state with a gap in the d_{xy} band while, for a large U/D , Δ_t causes a phase transition from a Hund’s metal to an insulating state with band (d_{xy}) + Mott ($d_{xz/yz}$) gaps.”

List of changes

1. **Figure 1c** is revised.

2. In the main text, 3rd paragraph,

“There have been previous experimental studies that demonstrated the manipulation of orbital polarization²²⁻²⁴. They include the charge transfer at cuprate-manganite interfaces²², metal-insulator transition in VO_2 ²³, and nickelate-cuprate heterostructures²⁴. However, the impact of J in such orbital polarization changes has not been considered. Here, our objective is to explore the interplay between Δ_t and J by controlling the orbital polarization in a widely accepted Hund’s system.”

3. In the main text, 5th paragraph,

“With $J = 0$, it can be challenging to detect the Mott transition induced by the crystal field because it only happens in a small range of U . On the other hand, J facilitates such transition in a much wider interaction range. Therefore, the system with a sizable J experiences orbital-selective phase transitions as Δ_t varies. For small U/D , increasing Δ_t induces a transition from a metallic state into another metallic state with a gap in the d_{xy} band while, for a large U/D , Δ_t causes a phase transition from a Hund’s metal to an insulating state with band (d_{xy}) + Mott ($d_{xz/yz}$) gaps.”

4. In the main text, 20th paragraph,

“Note that J and crystal field energy scales compete each other in terms of the orbital polarization. Therefore, the observed orbital-selective phase transition induced by the crystal field tuning in Hund’s system will provide an insight on the role of J .”

5. In supplementary information, **Supplementary Fig. 15** were added.